# SAFE OFFLINE REINFORCEMENT LEARNING WITH FEASIBILITY-GUIDED DIFFUSION MODEL

**Yinan Zheng**[1,2*]**, Jianxiong Li**[1,2*]**, Dongjie Yu**[3]**, Yujie Yang**[2]**, Shengbo Eben Li**[2]**,
Xianyuan Zhan**[1,4†] **, Jingjing Liu**[1,2†]
[1] Institute for AI Industry Research (AIR), Tsinghua University
[2] School of Vehicle and Mobility, Tsinghua University
[3] Department of Computer Science, The University of Hong Kong
[4] Shanghai Artificial Intelligence Laboratory
{zhengyn23,li-jx21}@mails.tsinghua.edu.cn,
zhanxianyuan@air.tsinghua.edu.cn

## ABSTRACT

Safe offline reinforcement learning is a promising way to bypass risky online interactions towards safe policy learning. Most existing methods only enforce soft constraints, *i.e.,* constraining safety violations in expectation below thresholds predetermined. This can lead to potentially unsafe outcomes, thus unacceptable in safety-critical scenarios. An alternative is to enforce the hard constraint of zero violation. However, this can be challenging in offline setting, as it needs to strike the right balance among three highly intricate and correlated aspects: safety constraint satisfaction, reward maximization, and behavior regularization imposed by offline datasets. Interestingly, we discover that via reachability analysis of safe-control theory, the hard safety constraint can be equivalently translated to identifying the largest feasible region given the offline dataset. This seamlessly converts the original trilogy problem to a feasibility-dependent objective, *i.e.*, maximizing reward value within the feasible region while minimizing safety risks in the infeasible region. Inspired by these, we propose *FISOR* (*FeasIbility-guided Safe Offline RL*), which allows safety constraint adherence, reward maximization, and offline policy learning to be realized via three decoupled processes, while offering strong safety performance and stability. In FISOR, the optimal policy for the translated optimization problem can be derived in a special form of weighted behavior cloning, which can be effectively extracted with a guided diffusion model thanks to its expressiveness. Moreover, we propose a novel energy-guided sampling method that does not require training a complicated time-dependent classifier to simplify the training. We compare FISOR against baselines on *DSRL* benchmark for safe offline RL. Evaluation results show that FISOR is the only method that can guarantee safety satisfaction in all tasks, while achieving top returns in most tasks. Project website: https://zhengyinan-air.github.io/FISOR/.

## 1 INTRODUCTION

Autonomous decision making imposes paramount safety concerns in safety-critical tasks (Li, 2023), such as industrial control systems (Zhan et al., 2022) and autonomous driving (Kiran et al., 2021), where any unsafe outcome can lead to severe consequences. In these tasks, one top priority is to ensure persistent safety constraint satisfaction (Zhao et al., 2023). Safe reinforcement learning (RL) holds the promise of safety guarantee by formulating and solving this problem as a Constrained Markov Decision Process (CMDP) (Gu et al., 2022). However, most previous studies focus on online RL setting, which suffers from serious safety issues in both training and deployment phases, especially for scenarios that lack high-fidelity simulators and require real system interaction for policy learning (Liu et al., 2023a). *Safe offline RL*, on the other hand, incorporates safety constraints into offline RL (Levine et al., 2020) and provides a new alternative for learning safe policies in a fully offline manner (Xu et al., 2022b). In this setting, only the learned final policy needs safety guar-

---

[*]Equal contribution
[†]Correspondence to Xianyuan Zhan and Jingjing Liu

antees, bypassing the inherent safety challenges during online training, thus providing an effective and practical solution to safety-critical real-world applications (Zhan et al., 2022).

Though promising, current safe offline RL methods exhibit several limitations. Firstly, to the best of our knowledge, all previous studies adopt *soft constraint* and only require constraint violations in expectation to remain under a given threshold throughout the entire trajectory (Xu et al., 2022b; Lee et al., 2022), thus allowing potential unsafe outcome. Such soft constraint-based formulation is widely employed in the research community but unacceptable to safety-critical industrial application (Yang et al., 2023a). Instead, *hard constraint* is more preferable for these tasks, where state-wise zero constraint violation is strictly enforced. To achieve this, a far more rigorous safety constraint is in demand, which however, inevitably imposes severe policy conservatism and suboptimality. Secondly, the difficulty is further exacerbated by the *offline setting*, as additional consideration of behavior regularization imposed by offline dataset is required to combat distributional shift (Fujimoto et al., 2019). Thus, striking the right balance among *constraint satisfaction*, *reward maximization*, and *offline policy learning* can be exceptionally challenging. Jointly optimizing these intricate and closely-correlated aspects, as in existing safe offline RL methods, can lead to very unstable training processes and unsatisfactory safety performance (Lee et al., 2022; Saxena & Cao, 2021).

To tackle these challenges, we introduce a novel safe offline RL approach, *FISOR (FeasIbility-guided Safe Offline RL)*, which provides stringent safety assurance while simultaneously optimizing for high rewards. FISOR revisits safety constraint satisfaction in view of optimization under different feasibility conditions and comprises three simple decoupled learning processes, offering strong safety performance and learning stability. Specifically, we first revise Hamilton-Jacobi (HJ) Reachability (Bansal et al., 2017) in safe-control theory to directly identify the largest feasible region through the offline dataset using a reversed version of expectile regression, which ensures zero constraint violation while maximally expanding the set of feasible solutions. With the knowledge of feasibility in hand, we further develop a feasibility-dependent objective that maximizes reward values within the feasible regions while minimizing safety risks in the infeasible regions. This allows constraint satisfaction and reward maximization to be executed in decoupled offline learning processes. Furthermore, we show that the optimal policy for these reformulated problems has a special weighted behavior cloning form with distinct weighting schemes in feasible and infeasible regions. By noting the inherent connection between weighted regression and exact energy-guided sampling for diffusion models (Lu et al., 2023), we extract the policy using a novel time-independent classifier-guided method, enjoying both superior expressivity and efficient training.

Extensive evaluations on the standard safe offline RL benchmark DSRL (Liu et al., 2023a) show that FISOR is the only method that guarantees satisfactory safety performance in all evaluated tasks, while achieving the highest returns in most tasks. Moreover, we demonstrate the versatility of FISOR in the context of safe offline imitation learning (IL), still outperforming competing baselines.

## 2 PRELIMINARY

Safe RL is typically formulated as a Constrained Markov Decision Process (Altman, 2021), which is specified by a tuple $\mathcal{M} := (\mathcal{S}, \mathcal{A}, T, r, h, c, \gamma)$. $\mathcal{S}$ and $\mathcal{A}$ represent the state and action space; $T : \mathcal{S} \times \mathcal{A} \to \Delta(\mathcal{S})$ is transition dynamics; $r : \mathcal{S} \times \mathcal{A} \to \mathbb{R}$ is the reward function; $h : \mathcal{S} \to \mathbb{R}$ is the constraint violation function; $c : \mathcal{S} \to [0, C_{\max}]$ is cost function; and $\gamma \in (0, 1)$ is the discount factor. Typically, $c(s) = \max(h(s), 0)$, which means that it takes on the value of $h(s)$ when the state constraint is violated ($h(s) > 0$), and zero otherwise ($h(s) \leq 0$).

Previous studies typically aim to find a policy $\pi : \mathcal{S} \to \Delta(\mathcal{A})$ to maximize the expected cumulative rewards while satisfying the soft constraint that restricts the expected cumulative costs below a pre-defined cost limit $l \geq 0$, *i.e.*, $\max_\pi \mathbb{E}_{\tau \sim \pi} \left[ \sum_{t=0}^\infty \gamma^t r(s_t, a_t) \right]$, s.t. $\mathbb{E}_{\tau \sim \pi} \left[ \sum_{t=0}^\infty \gamma^t c(s_t) \right] \leq l$, where $\tau$ is the trajectory induced by policy $\pi$. Existing safe offline RL methods typically solve this problem in the following form (Xu et al., 2022b; Lee et al., 2022):

$$\max_\pi \mathbb{E}_s \left[ V_r^\pi(s) \right] \qquad \text{s.t. } \mathbb{E}_s \left[ V_c^\pi(s) \right] \leq l; \ \mathrm{D}(\pi \| \pi_\beta) \leq \epsilon, \tag{1}$$

where $V_r^\pi$ is state-value function, $V_c^\pi$ is cost state-value function, $\pi_\beta$ is the underlying behavioral policy of the offline dataset $\mathcal{D} := (s, a, s', r, c)$ with both safe and unsafe trajectories. $\mathrm{D}(\pi \| \pi_\beta)$ is a divergence term (*e.g.*, KL divergence $\mathrm{D}_{\mathrm{KL}}(\pi \| \pi_\beta)$) used to prevent distributional shift from $\pi_\beta$ in offline setting. To handle safety constraints, previous methods often consider solving the Lagrangian

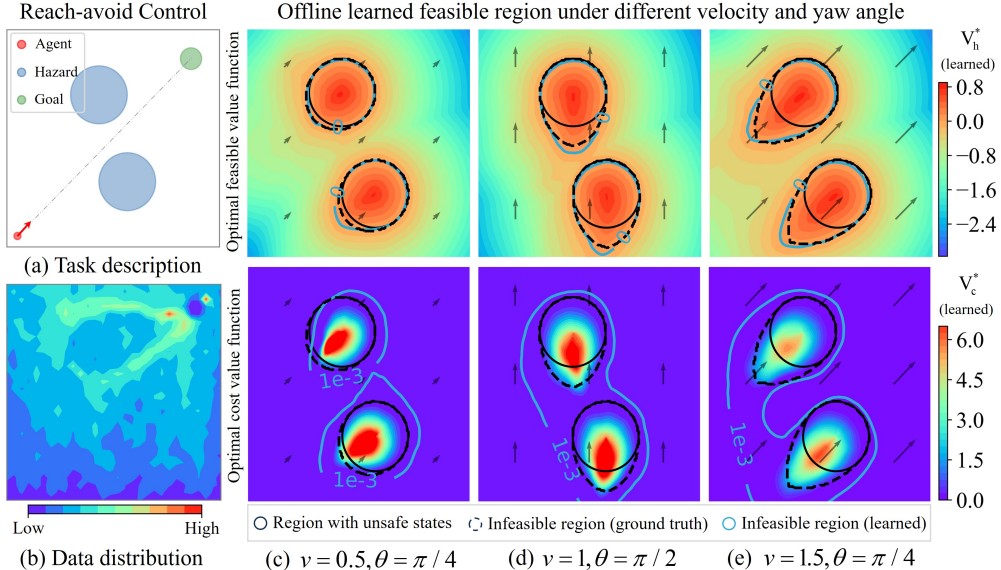

Figure 1: (a) Reach-avoid control task: the agent (red) aim to reach the goal (green) while avoiding hazards (blue). (b) Offline data distribution. (c)-(e) Comparisons with the feasible region learned by feasible value $\{s|V_h^*(s) \leq 0\}$ and cost value $\{s|V_c^*(s) \leq 1e^{-3}\}$. See Appendix D.1 for more details.

dual form of Eq. (1) with a Lagrangian multiplier $\lambda$ (Chow et al., 2017; Tessler et al., 2018):

$$\min_{\lambda \geq 0} \max_{\pi} \mathbb{E}_s \left[ V_r^\pi(s) - \lambda \left( V_c^\pi(s) - l \right) \right] \qquad \text{s.t. } D(\pi \| \pi_\beta) \leq \epsilon. \qquad (2)$$

**Limitations of Existing Methods**. The safety constraint in Eq. (1) is a *soft constraint* enforced on the expectation of all possible states, meaning that there exists a certain chance of violating the constraint with a positive $l$, which poses serious safety risks in real-world scenarios (Ma et al., 2022). Additionally, choosing a suitable cost limit $l$ that yields the best safety performance requires engineering insights, whereas the optimal limit may vary across tasks (Yu et al., 2022a). Besides, Eq. (1) shows that safe offline RL necessitates the simultaneous optimization of three potentially conflicting objectives: maximizing rewards, ensuring safety, and adhering to the policy constraint w.r.t the behavior policy. Finding the right balance across all three is very difficult. Moreover, the optimization objective in Eq. (2) introduces an additional layer of optimization for the Lagrange multiplier, and the learning of $V_r^\pi$, $V_c^\pi$ and $\pi$ are mutually coupled, where minor approximation errors can bootstrap across them and lead to severe instability (Kumar et al., 2019).

## 3 METHODS

**Feasibility-Dependent Optimization**. To address these challenges, we replace soft constraint with hard constraints that demand state-wise zero violations ($h(s_t) \leq 0, a \sim \pi, \forall t \in \mathbb{N}$, Appendix A.2). A straightforward way is to remove the expectation and set the cost limit to zero, i.e., replacing the safety constraint in Eq. (1) with $V_c^\pi \leq 0$:

$$\max_{\pi} \mathbb{E}_s \left[ V_r^\pi(s) \right] \qquad \text{s.t. } V_c^\pi(s) \leq 0; \ D(\pi \| \pi_\beta) \leq \epsilon. \qquad (3)$$

However, this leads to a highly restricted policy set $\Pi_c(s) := \{\pi | V_c^\pi(s) = 0\}$, since $V_c^\pi$ is non-negative ($c \geq 0$) and the presence of approximation errors makes it difficult to take the exact value of 0 (Figure 1). To address this issue, We replace $\Pi_c(s)$ with a new safe policy set $\Pi_f(s)$ suitable for practical solving. Furthermore, not all states allow the existence of a policy that satisfies the hard constraint, rendering the problem unsolvable at these states. We refer to these as *infeasible states* and the others *feasible states*. To solve this, we consider infeasible states separately and modify the problem as follows (Appendix B.1 shows theoretical relationships between Eq. (3) and Eq. (4)):

$$
\textbf{Feasible: } \max_{\pi} \mathbb{E}_s \left[ V_r^\pi(s) \cdot \mathbb{I}_{s \in \mathcal{S}_f} \right] \qquad \textbf{Infeasible: } \max_{\pi} \mathbb{E}_s \left[ -V_c^\pi(s) \cdot \mathbb{I}_{s \notin \mathcal{S}_f} \right]
$$
$$
\text{s.t. } \pi \in \Pi_f(s), \forall s \in \mathcal{S}_f \qquad\qquad \text{s.t. } D_{\text{KL}}(\pi \| \pi_\beta) \leq \epsilon \qquad\qquad (4)
$$
$$
D_{\text{KL}}(\pi \| \pi_\beta) \leq \epsilon
$$

Eq. (4) presents a feasibility-dependent objective. Here, $\mathbb{I}$ is the indicator function, and $\mathcal{S}_f$ is the feasible region including all feasible states, for which at least one policy exists that satisfies the hard constraint. This objective allows us to concentrate on maximizing rewards and minimizing safety risks individually to obtain the best possible performance. In the case of infeasible states, maximizing rewards becomes meaningless since safety violations will always occur. Therefore, the focus is shifted to minimizing future constraint violation as much as possible (Yu et al., 2022a). For feasible states, adherence to the hard constraint is ensured by policies from the safe policy set $\Pi_f$, thus allowing for the focus on reward maximization. Next, we introduce using the reachability analysis from safe-control theory to determine $\mathcal{S}_f$ and $\Pi_f$.

### 3.1 Offline Identification of Feasibility

Accurately determining $\mathcal{S}_f$ and $\Pi_f$ plays a crucial role in the success of the feasibility-dependent objective. This reminds us of the effectiveness of Hamilton-Jacobi (HJ) reachability (Bansal et al., 2017) for enforcing hard constraint in safe-control theory, which has recently been adopted in safe online RL studies (Fisac et al., 2019; Yu et al., 2022a). We first provide a brief overview of the basics in HJ reachability (Definition 1) and then delve into the instantiation of the feasible region $\mathcal{S}_f$ and policy set $\Pi_f$ in the HJ reachability framework (Definition 2-3).

**Definition 1** (Optimal feasible value function). *The optimal feasible state-value function $V_h^*$, and the optimal feasible action-value function $Q_h^*$ are defined as (Bansal et al., 2017):*

$$V_h^*(s) := \min_\pi V_h^\pi(s) := \min_\pi \max_{t\in\mathbb{N}} h(s_t), s_0 = s, a_t \sim \pi(\cdot \mid s_t), \tag{5}$$

$$Q_h^*(s,a) := \min_\pi Q_h^\pi(s,a) := \min_\pi \max_{t\in\mathbb{N}} h(s_t), s_0 = s, a_0 = a, a_{t+1} \sim \pi(\cdot \mid s_{t+1}), \tag{6}$$

where $V_h^\pi$ represents the maximum constraint violations in the trajectory induced by policy $\pi$ starting from a state $s$. The (optimal) feasible value function possesses the following properties:

- $V_h^\pi(s) \leq 0 \Rightarrow \forall s_t, h(s_t) \leq 0$, indicating $\pi$ can satisfy the hard constraint starting from $s$. Moreover, $V_h^*(s) \leq 0 \Rightarrow \min_\pi V_h^\pi \leq 0 \Rightarrow \exists \pi, V_h^\pi \leq 0$, meaning that there exists a policy that satisfies the hard constraint.

- $V_h^\pi(s) > 0 \Rightarrow \exists s_t, h(s_t) > 0$, suggesting $\pi$ cannot satisfy the hard constraint; and $V_h^*(s) > 0 \Rightarrow \min_\pi V_h^\pi > 0 \Rightarrow \forall \pi, V_h^\pi > 0$, meaning that no policy can satisfy the hard constraint.

We can see from these properties that the feasible value function indicates whether policies can satisfy hard constraints (*i.e.*, feasibility of policies); while the optimal feasible value function represents whether there exists a policy that achieves hard constraints (*i.e.*, feasibility of state), from which we define the feasible region (Definition 2) (Yu et al., 2022a) and feasible policy set (Definition 3):

**Definition 2** (Feasible region). *The feasible region of $\pi$ and the largest feasible region are:*

$$\mathcal{S}_f^\pi := \{s | V_h^\pi(s) \leq 0\} \qquad\qquad \mathcal{S}_f^* := \{s | V_h^*(s) \leq 0\}$$

**Definition 3** (Feasible policy set). *For state $s$, the feasible policy set is: $\Pi_f(s) := \{\pi | V_h^\pi(s) \leq 0\}$.*

$\mathcal{S}_f^\pi$ is induced by policy $\pi$ and includes all the states in which $\pi$ can satisfy the hard constraint. $\mathcal{S}_f^*$ is the largest feasible region, which is the union of all feasible regions (Choi et al., 2021; Li, 2023). Intuitively, $\mathcal{S}_f^*$ includes all the states where there exists at least one policy that satisfies hard constraint, and this property formally satisfies the requirement of feasible region in Eq. (4). Besides, $\Pi_f(s)$ includes all policies that satisfy hard constraint starting from state $s$. This set can be employed as a constraint in Eq. (4). Note that the range of $V_h$ is the entire real number space, making $V_h$ more suitable as a hard constraint compared to $V_c$, as shown in Figure 1.

**Learning Feasible Value Function from Offline Data**. Although HJ reachability holds the promise for enforcing hard constraint, calculating the optimal feasible value function based on Definition 1 requires Monte-Carlo estimation via interacting with the environment (Bajcsy et al., 2019), which is inaccessible in offline setting. Fortunately, similar to approximate dynamic programming, we can obtain the approximated optimal feasible value by repeatedly applying a feasible bellman operator with a discounted factor $\gamma \to 1$ (Fisac et al., 2019) (Definition 4).

**Definition 4** (Feasible Bellman operator). *The feasible Bellman operator defined below is a contraction mapping, with its fixed point $Q_{h,\gamma}^*$ satisfying $\lim_{\gamma\to1} Q_{h,\gamma}^* \to Q_h^*$ (Fisac et al., 2019).*

$$\mathcal{P}^* Q_h(s,a) := (1-\gamma)h(s) + \gamma \max\{h(s), V_h^*(s')\}, \quad V_h^*(s') = \min_{a'} Q_h(s',a') \tag{7}$$

Given the offline dataset $\mathcal{D}$, we can approximate $Q_h^*$ by minimizing $\mathbb{E}_{\mathcal{D}}[(\mathcal{P}^* Q_h - Q_h)^2]$. Note that the $\min_{a'}$ operation in Eq. (7) could query the out-of-distribution (OOD) actions, potentially leading to severe underestimation issues in offline setting (Fujimoto et al., 2019). To mitigate this, we consider a data support constrained operation $\min_{a', \text{s.t.} \pi_\beta(a'|s')>0}$ that aims to estimate the minimum value over actions that are in the support of data distribution, but this requires an estimation of the behavior policy $\pi_\beta$. Inspired by Kostrikov et al. (2022) that uses expectile regression to approximate the maximum value function without explicit behavioral modeling, we adopt a reversed version of this for learning the optimal (minimum) feasible value function via minimizing Eq. (8)-(9):

$$\mathcal{L}_{V_h} = \mathbb{E}_{(s,a)\sim\mathcal{D}}\left[L_{\text{rev}}^\tau\left(Q_h(s,a) - V_h(s)\right)\right], \tag{8}$$

$$\mathcal{L}_{Q_h} = \mathbb{E}_{(s,a,s')\sim\mathcal{D}}\left[\left(((1-\gamma)h(s) + \gamma\max\{h(s), V_h(s')\}) - Q_h(s,a)\right)^2\right], \tag{9}$$

where $L_{\text{rev}}^\tau(u) = |\tau - \mathbb{I}(u > 0)|u^2$. For $\tau \in (0.5, 1)$, this asymmetric loss diminishes the impact of $Q_h$ values that exceed $V_h$, placing greater emphasis on smaller values instead. Subsequently, we can substitute $V_h(s')$ for $\min_{a'} Q_h(s', a')$ within the feasible Bellman operator. By minimizing Eq. (8) and Eq. (9), we can predetermine the approximated largest feasible region, as shown in Figure 1.

## 3.2 FEASIBILITY-DEPENDENT OPTIMIZATION OBJECTIVE

Given the feasible policy set $\Pi_f$ and the predetermined largest feasible region $\mathcal{S}_f^*$, we can instantiate the feasibility-dependent objective in Eq. (4) as follows, with $V_c$ changed to $V_h$:

$$\textbf{Feasible: } \max_\pi \mathbb{E}_s\left[V_r^\pi(s) \cdot \mathbb{I}_{s \in \mathcal{S}_f^*}\right] \quad \textbf{Infeasible: } \max_\pi \mathbb{E}_s\left[-V_h^\pi(s) \cdot \mathbb{I}_{s \notin \mathcal{S}_f^*}\right]$$
$$\text{s.t. } V_h^\pi(s) \le 0, \forall s \in \mathcal{S}_f^* \qquad\qquad \text{s.t. } \text{D}_{\text{KL}}(\pi\|\pi_\beta) \le \epsilon \tag{10}$$
$$\text{D}_{\text{KL}}(\pi\|\pi_\beta) \le \epsilon$$

Eq. (10) is similar to the objective in the safe online RL method, RCRL (Yu et al., 2022a). However, RCRL suffers from severe training instability due to its coupling training procedure. In contrast, we predetermine $\mathcal{S}_f^*$ from the offline data. However, the training of $V_r^\pi$, $V_h^\pi$, and $\pi$ is still coupling together, leading to potential training instability. To solve this, we show in Lemma 1 that these interrelated elements can be fully decoupled. Based on this insight, we devise a new surrogate learning objective that disentangles the intricate dependencies among these elements.

**Lemma 1.** *The optimization objectives and safety constraints in Eq. (10) can be achieved by separate optimization objectives and constraints as follows (see Appendix B.2 for proof):*

- *Optimization objective in the feasible region:* $\max_\pi \mathbb{E}_{a\sim\pi}\left[A_r^*(s,a)\right] \Rightarrow \max_\pi \mathbb{E}_s\left[V_r^\pi(s)\right].$

- *Optimization objective in the infeasible region:* $\max_\pi \mathbb{E}_{a\sim\pi}\left[-A_h^*(s,a)\right] \Rightarrow \max_\pi \mathbb{E}_s\left[-V_h^\pi(s)\right].$

- *Safety constraint in the feasible region:* $\int_{\{a|Q_h^*(s,a)\le 0\}} \pi(\cdot|s)\mathrm{d}a = 1 \Rightarrow V_h^\pi(s) \le 0.$

Lemma 1 shows that the maximization objectives in Eq. (10) can be achieved by finding the optimal advantages $A_r^*$, $A_h^*$ first, and then optimizing the policy $\pi$. The safety constraints in Eq. (10) can also be enforced by using the predetermined $Q_h^*$ without coupling to other networks. Inspired by this, we propose a surrogate objective in Eq. (11) that fully decouples the intricate interrelations between the optimizations of $V_r^\pi$, $V_h^\pi$ and $\pi$. We dub this *FISOR (FeasIbility-guided Safe Offline RL)* :

$$\textbf{Feasible: } \max_\pi \mathbb{E}_{a\sim\pi}\left[A_r^*(s,a) \cdot \mathbb{I}_{V_h^*(s)\le 0}\right] \quad \textbf{Infeasible: } \max_\pi \mathbb{E}_{a\sim\pi}\left[-A_h^*(s,a) \cdot \mathbb{I}_{V_h^*(s)>0}\right]$$
$$\text{s.t. } \int_{\{a|Q_h^*(s,a)\le 0\}} \pi(a|s)\mathrm{d}a = 1, \forall s \in \mathcal{S}_f^* \qquad \text{s.t. } \int_a \pi(\cdot|s)\mathrm{d}a = 1 \tag{11}$$
$$\text{D}_{\text{KL}}(\pi\|\pi_\beta) \le \epsilon \qquad\qquad\qquad \text{D}_{\text{KL}}(\pi\|\pi_\beta) \le \epsilon$$

We present an intuitive illustration in Figure 2 to demonstrate the characteristics of Eq. (11). In feasible regions, FISOR favors actions that yield high rewards while ensuring that the agent remains within feasible regions. The largest feasible region grants the policy with more freedom to maximize reward, allowing for enhanced performance. As depicted in Figure 2, agents originating from feasible regions can effectively find a direct and safe path to the goal.

In infeasible regions, FISOR chooses actions that minimize constraint violations as much as possible. Figure 2 shows that the agents starting from infeasible regions initially prioritize escaping from infeasible regions, and then navigate towards the destination along safe paths.

The decoupled optimization objective in Eq. (11) allows us to directly obtain the closed-form solution as shown in Theorem 1 below (see Appendix B.3 for detailed proof):

**Theorem 1.** *The optimal solution for Eq. (11) satisfies:*

$$\pi^*(a|s) \propto \pi_\beta(a|s) \cdot w(s,a), \tag{12}$$

*where $w(s,a)$ is the weight function, and $\alpha_1$, $\alpha_2$ are temperatures that control the behavior regularization strength:*

$$w(s,a) = \begin{cases} \exp\left(\alpha_1 A_r^*(s,a)\right) \cdot \mathbb{I}_{Q_h^*(s,a) \leq 0} & V_h^*(s) \leq 0 \\ \exp\left(-\alpha_2 A_h^*(s,a)\right) & V_h^*(s) > 0 \end{cases} \tag{13}$$

The optimal solution in Eq. (13) is similar to weighted behavior cloning, but adopts distinct weighting schemes in feasible and infeasible regions. Using Eq. (13), we can easily extract the optimal policy in a remarkably simple and stable way, while achieving a balance between safety assurance, reward maximization, and behavior regularization.

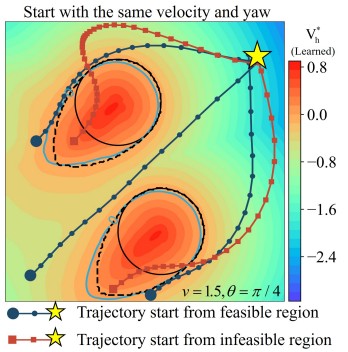

Figure 2: Trajectories induced by FISOR from different start points.

### 3.3 GUIDED DIFFUSION POLICY LEARNING WITHOUT TIME-DEPENDENT CLASSIFIER

Theorem 1 shows that the optimal policy $\pi^*$ needs to choose drastically different behaviors based on the current state's feasibility. We use diffusion model to parameterize policies due to its strong ability in modeling complex distributions (Ho et al., 2020; Wang et al., 2023c). To obtain the weighted distribution of $\pi^*$ in Eq. (12), a simple way is to use classifier-guidance methods to guide the diffusion sampling procedures (Lin et al., 2023; Lu et al., 2023). However, such methods require training an additional complicated time-dependent classifier, inevitably increasing the overall complexity. Instead, we demonstrate in Theorem 2 that the exact $\pi^*$ in Eq. (12) can in fact be obtained without the training of time-dependent classifier, thus greatly reducing training difficulty.

**Theorem 2** (Weighted regression as exact energy guidance). *We can sample $a \sim \pi^*(a|s)$ by optimizing the weighted regression loss in Eq. (14) and solving the diffusion ODEs/SDEs given the obtained $z_\theta$. (see Appendix C for proof and thorough discussions)*

Specifically, the exact $\pi^*$ can be obtained by modifying the diffusion training loss for the behavior policy $\pi_\beta$: $\min_\theta \mathbb{E}_{t,(s,a),z}\left[\|z - z_\theta(a_t,s,t)\|_2^2\right]$ via simply augmenting the weight function $w(s,a)$ in Eq. (13) with the following weighted loss:

$$\min_\theta \mathcal{L}_z(\theta) = \min_\theta \mathbb{E}_{t\sim\mathcal{U}([0,T]),z\sim\mathcal{N}(0,I),(s,a)\sim\mathcal{D}}\left[w(s,a) \cdot \|z - z_\theta(a_t,s,t)\|_2^2\right], \tag{14}$$

where $a_t = \alpha_t a + \sigma_t z$ is the noisy action that satisfies the forward transition distribution $\mathcal{N}(a_t|\alpha_t a, \sigma_t^2 I)$ in the diffusion model, and $\alpha_t, \sigma_t$ are human designed noise schedules. After obtaining $\theta$, we can sample from the approximated optimal policy $a \sim \pi_\theta^*(a|s)$ by solving the diffusion ODEs/SDEs (Song et al., 2021) as stated in Theorem 2. Eq. (14) bypasses the training of the complicated time-dependent classifier in typical guided diffusion models (Lin et al., 2023; Lu et al., 2023) and forms a weighted regression objective, which has proved stable against directly maximizing the value functions (Kostrikov et al., 2022). This weighted loss form has been used in recent studies (Hansen-Estruch et al., 2023; Kang et al., 2023), but they did not provide theoretical investigation in its inherent connections to exact energy guidance for diffusion policies (Lu et al., 2023) (see Appendix C.3 for comparison with other guided sampling methods).

### 3.4 PRACTICAL IMPLEMENTATION

Theorem 1 offers a decoupled solution for extracting the optimal policy from the dataset. The remaining challenge is how to obtain the optimal advantages $A_r^*$ in advance. In-sample learning methods, such as IQL (Kostrikov et al., 2022), SQL (Xu et al., 2023) and XQL (Garg et al., 2023), decouple policy learning from value function learning and allows the approximation of $A_r^* = Q_r^* - V_r^*$ without explicit policy, offering improved training stability. We use IQL to learn the optimal value function $Q_r^*$ and $V_r^*$ in our method with $L^\tau(u) = |\tau - \mathbb{I}(u < 0)| u^2, \tau \in (0.5, 1)$:

$$\mathcal{L}_{V_r} = \mathbb{E}_{(s,a)\sim\mathcal{D}}\left[L^\tau\left(Q_r(s,a) - V_r(s)\right)\right], \tag{15}$$

$$\mathcal{L}_{Q_r} = \mathbb{E}_{(s,a,s')\sim\mathcal{D}}\left[\left(r + \gamma V_r(s') - Q_r(s,a)\right)^2\right]. \tag{16}$$

Table 1: Normalized DSRL (Liu et al., 2023a) benchmark results. ↑ means the higher the better. ↓ means the lower the better. Each value is averaged over 20 evaluation episodes and 3 seeds. Gray: Unsafe agents. **Bold**: Safe agents whose normalized cost is smaller than 1. **Blue**: Safe agents with the highest reward.

| Task | BC | | CDT | | BCQ-Lag | | CPQ | | COptiDICE | | TREBI | | FISOR (ours) | |
|---|---|---|---|---|---|---|---|---|---|---|---|---|---|---|
| | reward ↑ | cost ↓ | reward ↑ | cost ↓ | reward ↑ | cost ↓ | reward ↑ | cost ↓ | reward ↑ | cost ↓ | reward ↑ | cost ↓ | reward ↑ | cost ↓ |
| CarButton1 | 0.01 | 6.19 | 0.17 | 7.05 | 0.13 | 6.68 | 0.22 | 40.06 | -0.16 | 4.63 | 0.07 | 3.75 | **-0.02** | **0.26** |
| CarButton2 | -0.10 | 4.47 | 0.23 | 12.87 | -0.04 | 4.43 | 0.08 | 19.03 | -0.17 | 3.40 | -0.03 | 0.97 | **0.01** | **0.58** |
| CarPush1 | 0.21 | 1.97 | 0.27 | 2.12 | 0.23 | 1.33 | **0.08** | **0.77** | 0.21 | 1.28 | 0.26 | 1.03 | **0.28** | **0.28** |
| CarPush2 | 0.11 | 3.89 | 0.16 | 4.60 | 0.10 | 2.78 | -0.03 | 10.00 | 0.10 | 4.55 | 0.12 | 2.65 | **0.14** | **0.89** |
| CarGoal1 | 0.35 | 1.54 | 0.60 | 3.15 | 0.44 | 2.76 | 0.33 | 4.93 | 0.43 | 2.81 | 0.41 | 1.16 | **0.49** | **0.83** |
| CarGoal2 | 0.22 | 3.30 | 0.45 | 6.05 | 0.34 | 4.72 | 0.10 | 6.31 | 0.19 | 2.83 | 0.13 | 1.16 | **0.06** | **0.33** |
| AntVel | 0.99 | 12.19 | **0.98** | **0.91** | 0.85 | 18.54 | -1.01 | 0.00 | 1.00 | 10.29 | 0.31 | 0.00 | **0.89** | **0.00** |
| HalfCheetahVel | 0.97 | 17.93 | **0.97** | **0.55** | 1.04 | 57.06 | 0.08 | 2.56 | 0.43 | 0.00 | 0.87 | 0.23 | **0.89** | **0.00** |
| SwimmerVel | 0.38 | 2.98 | 0.67 | 1.47 | 0.29 | 4.10 | 0.31 | 11.58 | 0.58 | 23.64 | 0.42 | 1.31 | **-0.04** | **0.00** |
| **SafetyGym Average** | 0.35 | 6.05 | 0.50 | 4.31 | 0.38 | 11.38 | 0.02 | 10.58 | 0.29 | 5.94 | 0.28 | 1.36 | **0.30** | **0.35** |
| AntRun | 0.73 | 11.73 | 0.70 | 1.88 | 0.65 | 3.30 | **0.00** | **0.00** | 0.62 | 3.64 | 0.63 | 5.43 | **0.45** | **0.03** |
| BallRun | 0.67 | 11.38 | **0.32** | **0.45** | 0.43 | 6.25 | 0.85 | 13.67 | 0.55 | 11.32 | 0.29 | 4.24 | **0.18** | **0.00** |
| CarRun | 0.96 | 1.88 | 0.99 | 1.10 | 0.84 | 2.51 | 1.06 | 10.49 | **0.92** | **0.00** | 0.97 | 1.01 | 0.73 | 0.14 |
| DroneRun | 0.55 | 5.21 | **0.58** | **0.30** | 0.80 | 17.98 | 0.02 | 7.95 | 0.72 | 13.77 | 0.59 | 1.41 | **0.30** | **0.55** |
| AntCircle | 0.65 | 19.45 | 0.48 | 7.44 | 0.67 | 19.13 | **0.00** | **0.00** | 0.18 | 13.41 | 0.37 | 2.50 | **0.20** | **0.00** |
| BallCircle | 0.72 | 10.02 | 0.68 | 2.10 | 0.67 | 8.50 | 0.40 | 4.37 | 0.70 | 9.06 | 0.63 | 1.89 | **0.34** | **0.00** |
| CarCircle | 0.65 | 11.16 | 0.71 | 2.19 | 0.68 | 8.84 | 0.49 | 4.48 | 0.44 | 7.73 | **0.49** | **0.73** | 0.40 | 0.11 |
| DroneCircle | 0.82 | 13.78 | 0.55 | 1.29 | 0.95 | 18.56 | -0.27 | 1.29 | 0.24 | 2.19 | 0.54 | 2.36 | **0.48** | **0.00** |
| **BulletGym Average** | 0.72 | 10.58 | 0.63 | 2.09 | 0.71 | 10.63 | 0.32 | 5.28 | 0.55 | 7.64 | 0.56 | 2.45 | **0.39** | **0.10** |
| easysparse | 0.32 | 4.73 | **0.05** | **0.10** | 0.99 | 14.00 | **-0.06** | **0.24** | 0.94 | 18.21 | 0.26 | 6.22 | **0.38** | **0.53** |
| easymean | 0.22 | 2.68 | **0.27** | **0.24** | 0.54 | 10.35 | **-0.06** | **0.24** | 0.74 | 14.81 | 0.19 | 4.85 | **0.38** | **0.25** |
| easydense | 0.20 | 1.70 | 0.43 | 2.31 | 0.40 | 6.64 | **-0.06** | **0.29** | 0.60 | 11.27 | 0.26 | 5.81 | **0.36** | **0.25** |
| mediumsparse | 0.53 | 1.74 | 0.26 | 2.20 | 0.93 | 7.48 | **-0.08** | **0.18** | 0.64 | 7.26 | 0.06 | 1.70 | **0.42** | **0.22** |
| mediummean | 0.66 | 2.94 | 0.28 | 2.13 | 0.60 | 6.35 | **-0.08** | **0.28** | 0.73 | 8.35 | 0.20 | 1.90 | **0.39** | **0.08** |
| mediumdense | 0.65 | 3.79 | **0.29** | **0.77** | 0.64 | 3.78 | **-0.08** | **0.20** | 0.91 | 9.52 | 0.03 | 1.18 | **0.49** | **0.44** |
| hardsparse | 0.28 | 1.98 | **0.17** | **0.47** | 0.48 | 7.52 | **-0.04** | **0.28** | 0.34 | 7.34 | **0.00** | **0.82** | **0.30** | **0.01** |
| hardmean | 0.34 | 3.76 | 0.28 | 3.32 | 0.31 | 5.11 | **-0.05** | **0.24** | 0.36 | 7.51 | 0.16 | 4.91 | **0.26** | **0.09** |
| harddense | 0.40 | 5.57 | 0.24 | 1.49 | 0.39 | 5.11 | **-0.04** | **0.24** | 0.42 | 8.11 | 0.02 | 1.21 | **0.30** | **0.34** |
| **MetaDrive Average** | 0.40 | 3.21 | 0.25 | 1.45 | 0.59 | 7.48 | **-0.06** | **0.24** | 0.63 | 10.26 | 0.13 | 3.18 | **0.36** | **0.25** |

So far, we have transformed the original tightly-coupled safety-constrained offline RL problem into three decoupled simple supervised objectives: 1) Offline identification of the largest feasible region (Eq. (8-9)); 2) Optimal advantage learning (Eq. (15-16)); and 3) Optimal policy extraction via guided diffusion model (Eq. (14)). This disentanglement of learning objectives provides a highly stable and conducive solution to safe offline RL for practical applications. To further enhance safety, we sample $N$ action candidates from the diffusion policy and select the safest one (*i.e.*, the lowest $Q_h^*$ value) as the final output as in other safe RL methods (Dalal et al., 2018; Thananjeyan et al., 2021). Please see Appendix D.3 and D.4 for implementation details and algorithm pseudocode. Code is available at: `https://github.com/ZhengYinan-AIR/FISOR`.

## 4 EXPERIMENTS

**Evaluation Setups.** We conduct extensive evaluations on Safety-Gymnasium (Ray et al., 2019; Ji et al., 2023), Bullet-Safety-Gym (Gronauer, 2022) and MetaDrive (Li et al., 2022) tasks on DSRL benchmark (Liu et al., 2023a) to evaluate FISOR against other SOTA safe offline RL methods. We use *normalized return* and *normalized cost* as the evaluation metrics, where a normalized cost below 1 indicates safety. As stated in DSRL, we take safety as the primary criterion for evaluations, and pursue higher rewards on the basis of meeting safety requirements. In order to mimic safety-critical application scenarios, we set more stringent safety requirements, where the more difficult task Safety-Gymnasium has its cost limit set to 10, and other environments are set to 5.

**Baselines.** We compare FISOR with the following baselines: 1) *BC*: Behavior cloning that imitates the whole datasets. 2) *CDT* (Liu et al., 2023b): A Decision Transformer based method that considers safety constraints. 3) *BCQ-Lag*: A Lagrangian-based method that considers safety constraints on the basis of BCQ (Fujimoto et al., 2019). 4) *CPQ* (Xu et al., 2022b): Treat the OOD action as an unsafe action and update the Q-value function with safe state-action. 5) *COptiDICE* (Lee et al., 2022): A DICE (distribution correction estimation) based safe offline RL method that builds on OptiDICE (Lee et al., 2021). 6) *TREBI* (Lin et al., 2023): A diffusion-based method using classifier-guidance to sample safe trajectory. See Appendix D for more experimental details.

**Main Results.** Evaluation results are presented in Table 1. FISOR is the only method that achieves satisfactory safety performance in all tasks and obtains the highest return in most tasks, demonstrating its superiority in achieving both safety and high rewards. Other methods either suffer from

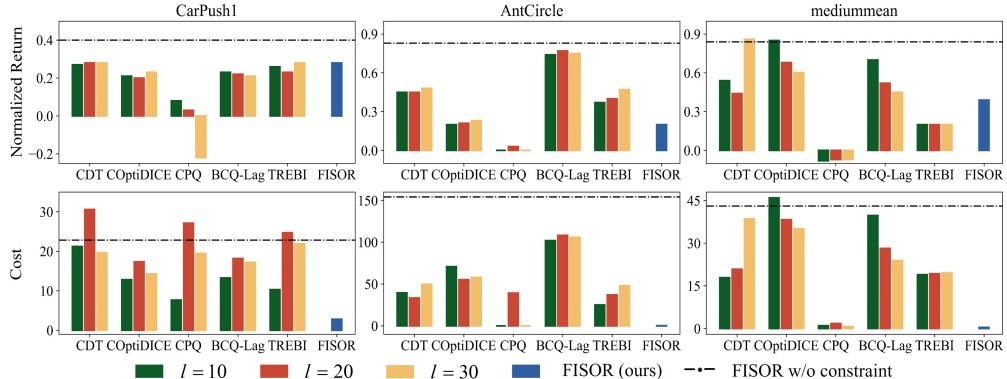

Figure 3: Soft constraint sensitivity experiments for cost limit $l$ in three environments.

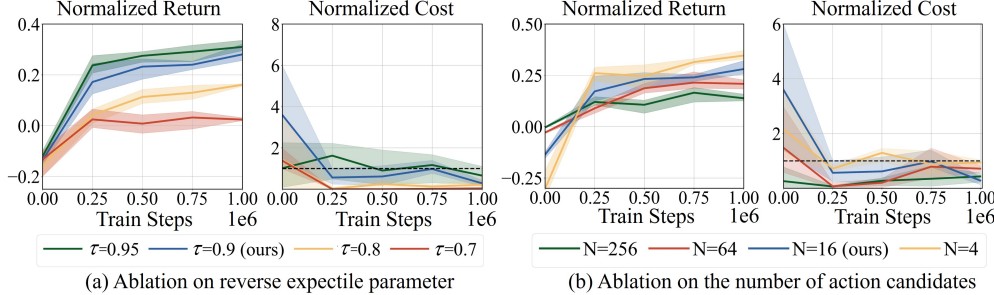

(a) Ablation on reverse expectile parameter     (b) Ablation on the number of action candidates

Figure 4: Ablations on hyperparameter choices in CarPush1.

severe constraint violations or suboptimal returns. *BCQ-Lag* tries to balance performance and safety using the Lagrangian method, but the coupled training procedure makes it unstable and difficult to find the right balance, leading to inferior safety and reward results. *CPQ* takes a more conservative approach by only updating the value function on safe state-action pairs, which can result in very low returns when constraints are enforced. *CDT* and *TREBI* use complex network architectures, which can meet the constraints and achieve high returns on some tasks, but their overall safety performance is not very impressive since they use soft constraints.

## 4.1 ABLATION STUDY AND ANALYSIS

**Soft Constraint Under Different Cost Limits**. We evaluate the sensitivity of cost limit selection $l$ for soft-constraint-based methods to demonstrate the effectiveness of hard constraint. We evaluate three cost limits and present the results in Figure 3. *FISOR w/o constraint* refers to our algorithm without safety constraints, focusing solely on maximizing rewards. Figure 3 shows that most soft-constraint-based methods are highly sensitive to the value of cost limit. In some cases, choosing a small cost limit even leads to an increase in the final cost. This shows that it is difficult for these methods to select the right cost limit to achieve the best performance, which requires task-specific tuning. In contrast, our algorithm, which considers hard constraints, does not encounter this issue and achieves superior results using only one group of hyperparameters.

**Hyperparameter Choices**. We sweep two hyperparameters: the expectile value $\tau$ and the number of sampled action candidates $N$ during evaluation. Figure 4 shows that $\tau$ and $N$ primarily affect conservatism level, but do not significantly impact safety satisfaction. A larger $\tau$ and smaller $N$ result in a more aggressive policy, whereas a smaller $\tau$ and larger $N$ lead to a more conservative behavior. We find that setting $\tau = 0.9$ and $N = 16$ consistently yields good results in terms of both safety and returns. Hence, we evaluate FISOR on all 26 tasks using these same hyperparameters. Please refer to Appendix D.5 for detailed hyperparameter setups.

**Ablations on Each Design Choice.** We demonstrate the effectiveness of individual components of our method: HJ reachability, feasibility-dependent objective, and diffusion model. 1) To verify the advantage of

Table 2: Ablations on HJ reachability.

| Task | w/o HJ | | FISOR | |
|---|---|---|---|---|
| | reward ↑ | cost ↓ | reward ↑ | cost ↓ |
| AntRun | 0.30 | 0.44 | 0.45 | 0.03 |
| BallRun | 0.08 | 0.14 | 0.18 | 0.00 |
| CarRun | -0.33 | 0.00 | 0.73 | 0.14 |
| DroneRun | -0.11 | 5.12 | 0.30 | 0.55 |
| AntCircle | 0.00 | 0.00 | 0.20 | 0.00 |
| BallCircle | 0.02 | 0.81 | 0.34 | 0.00 |
| CarCircle | 0.01 | 2.16 | 0.40 | 0.11 |
| DroneCircle | -0.21 | 0.26 | 0.48 | 0.00 |

Table 3: Ablations on infeasible objective and diffusion policies (normalized cost).

| Task | w/o infeasible | w/o diffusion | FISOR |
|------|----------------|---------------|-------|
| CarButton1 | 4.61 | 0.72 | 0.26 |
| CarButton2 | 6.53 | 1.73 | 0.58 |
| CarPush1 | 0.86 | 1.21 | 0.28 |
| CarPush2 | 3.10 | 4.45 | 0.89 |
| CarGoal1 | 4.00 | 2.71 | 0.83 |
| CarGoal2 | 5.46 | 2.82 | 0.33 |

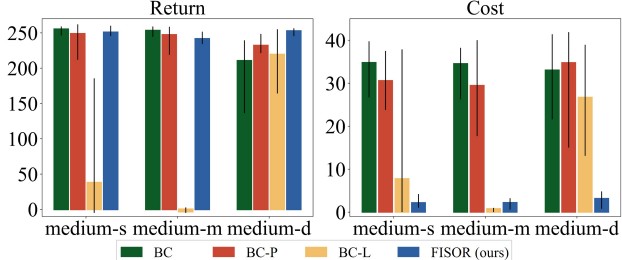

Figure 5: Safe offline IL results in MetaDrive.

using HJ reachability, we ablate on the choice of determining the largest feasible region using cost value function instead of feasible value function. This variant is denoted as *w/o HJ*. The results are summarized in Figure 1 and Table 2. For the variant *w/o HJ*, the presence of approximation errors renders the estimated infeasible region pretty large and inaccurate, which leads to severe policy conservatism and hurt final returns. By contrast, the feasible value function in FISOR can accurately identify the largest feasible region, which greatly increases optimized returns. 2) We train FISOR by setting the weight for the infeasible region in Eq. (13) to 0 (*w/o infeasible*) to examine the necessity of minimizing constraint violation in infeasible regions. 3) We also ablate on replacing diffusion policy with conventional Gaussian policy (*w/o diffusion*). The results are presented in Table 3, where *w/o infeasible* suffers from severe constraint violations without learning recovery behavior in infeasible regions. Meanwhile, *w/o diffusion* struggles to effectively fit the data distribution both inside and outside the feasible region, resulting in high constraint violations.

**Extension to Safe Offline Imitation Learning.** We also show the versatility of FISOR by extending it to the safe offline Imitation Learning (IL) setting, where we aim to learn a safe policy given both safe and unsafe high-reward trajectories without reward labels. In comparison to 1) BC, 2) BC with fixed safety penalty (*BC-P*), and 3) Lagrangian penalty (*BC-L*), FISOR can imitate expert behavior and meanwhile avoid unsafe outcomes, as illustrated in Figure 5. See Appendix D.6 for details.

## 5    RELATED WORK

Most existing safe RL studies focus on the online setting, which typically use the Lagrangian method to solve the constrained problem (Chow et al., 2017; Tessler et al., 2018; Ding et al., 2020), thus resulting in an intertwined problem with stability issues (Saxena & Cao, 2021). CPO (Achiam et al., 2017) utilizes trust region technique and theoretically guarantees safety during training. However, all these methods cannot guarantee safety during the entire online training phase. By contrast, safe offline RL learns policies from offline datasets without risky online interactions. CPQ (Xu et al., 2022b) is the first practical safe offline RL method that assigns high-cost values to both OOD and unsafe actions, which distorts the value function and may cause poor generalizability (Li et al., 2023b). COptiDICE (Lee et al., 2022) is a DICE-based method (Lee et al., 2021) with safety constraints, but the residual learning of DICE may lead to inferior results (Baird, 1995; Mao et al., 2023). Recently, some methods (Liu et al., 2023b; Lin et al., 2023) incorporate safety into Decision Transformer (Chen et al., 2021) or Diffuser (Janner et al., 2022) architecture, but are highly computational inefficient. Moreover, all these safe offline RL methods only consider soft constraint that enforces constraint violations in expectation, which lacks strict safety assurance. Some studies (Choi et al., 2020; Yang et al., 2023b; Wabersich et al., 2023) utilize safety certificates from control theory, such as control barrier function (CBF) (Ames et al., 2019; Lee et al., 2023; Kim et al., 2023) and HJ reachability (Bansal et al., 2017; Fisac et al., 2019; Yu et al., 2022a), to ensure state-wise zero violations (hard constraint), but only applicable to online RL.

## 6    CONCLUSION AND FUTURE WORK

We propose FISOR, which enables safe offline policy learning with hard safety constraints. We introduce an offline version of the optimal feasible value function in HJ reachability to characterize the feasibility of states.This leads to a feasibility-dependent objective which can be further solved using three simple decoupled learning objectives. Experiments show that FISOR can achieve superior performance and stability while guaranteeing safety requirements. It can also be extended to safe offline IL problems. One caveat is that limited offline data size could hurt the algorithm's performance. Therefore, extending FISOR to a safe offline-to-online RL framework can be a viable future direction to improve performance with online interactions, while retaining safety.

ACKNOWLEDGEMENT

This work is supported by National Key Research and Development Program of China under Grant (2022YFB2502904), and funding from Haomo.AI.

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

# A EXTENDED DISCUSSIONS ON RELATED WORKS

## A.1 OFFLINE RL

To address the issue of distributional shift when learning RL policies solely from offline datasets, a straightforward approach is to include policy constraints that enforce the learned policy to remain close to the behavior policy (Fujimoto et al., 2019; Kumar et al., 2019; Fujimoto & Gu, 2021; Li et al., 2023a) or generalizable regions in data (Li et al., 2023b; Cheng et al., 2023). Alternatively, value regularization methods do not impose direct policy constraints; instead, they penalize the value of OOD actions to mitigate distributional shift (Kumar et al., 2020; Kostrikov et al., 2021; Niu et al., 2022; Yang et al., 2022; Lyu et al., 2022). These methods require jointly training actor and critic networks, which can be unstable in practice. Recently, in-sample learning methods (Kostrikov et al., 2022; Xu et al., 2023; Wang et al., 2023a; Garg et al., 2023; Xu et al., 2022a; Ma et al., 2021b) exclusively leverage state-action pairs from the dataset for value and policy learning with decoupled losses, greatly enhancing stability. In addition, some recent studies employ expressive diffusion models as policies to capture complex distributions (Janner et al., 2022; Lu et al., 2023; Hansen-Estruch et al., 2023), also achieving impressive performances.

## A.2 ADDITIONAL DISCUSSION ON HARD CONSTRAINT AND SOFT CONSTRAINT

Based on different safety requirements, we divide existing works into two categories: soft constraint and hard constraint.

**Hard Constraint**. The hard constraint requires the policy to achieve state-wise zero constraint violation:

$$h(s_t) \leq 0, a \sim \pi, \forall t \in \mathbb{N}.$$

It can also be represented by cost function:

$$c(s_t) = 0, a \sim \pi, \forall t \in \mathbb{N}.$$

**Soft Constraint**. The most commonly used soft constraint restricts the expected cumulative costs below the cost limit $l$:

$$\mathbb{E}_{\tau \sim \pi} \left[ \sum_{t=0}^{\infty} c(s_t) \right] \leq l, \mathbb{E}_{\tau \sim \pi} \left[ \sum_{t=0}^{\infty} \gamma^t c(s_t) \right] \leq l.$$

There are a bunch of works focus on the state-wise constraint (Ma et al., 2021a; Zhao et al., 2023):

$$c(s_t) \leq w, a \sim \pi, \forall t \in \mathbb{N}.$$

Since this type of problem still allows for a certain degree of constraint violation, we classify it as a soft constraint.

**Discussions on Related Works (Hard Constraint).** There is a group of safe online RL works considering hard constraints that have inspired us. Wang et al. (2023b) proposed a method for learning safe policy with joint soft barrier function learning, generative modeling, and policy optimization. However, the intertwined learning process is not suitable for offline scenarios. SNO-MDP (Wachi & Sui, 2020) primarily focuses on the expansion of safe regions in online exploration. Additionally, the Gaussian process modeling approach may lead to considerable computational expense. In contrast, FISOR can learn the feasible region directly from the dataset without policy evaluation. ISSA (Zhao et al., 2021) can achieve zero constraint violations, but its need for task-specific prior knowledge to design safety constraints makes it unsuitable for general cases. RCRL (Yu et al., 2022a) and RE-SPO (Ganai et al., 2024) employ reachability theory in an online setting, but their use of Lagrangian iterative solving methods hinders algorithm stability. FISOR achieves stable training across multiple tasks through its decoupled optimization objectives.

**Algorithm Classification** In order to better distinguish the safety constraints used by existing methods and consider their training methods (offline or online), we summarize them as shown in Table 4. Many safe-control works pretrain safety certificates using offline methods (Robey et al., 2020; Zhao et al., 2021), but use them for online control without an offline trained policy. These methods are not considered as safe offline RL algorithms.

Table 4: Detailed comparisons with related safe RL methods.

|  | Online RL | Offline RL |
|---|---|---|
| Soft Constraint | CPO (Achiam et al., 2017)
RCRL (Chow et al., 2017)
RCPO (Tessler et al., 2018)
FOCOPS (Zhang et al., 2020)
PCPO (Yang et al., 2020)
Sauté (Sootla et al., 2022) | CBPL (Le et al., 2019)
CPQ (Xu et al., 2022b)
COptiDICE (Lee et al., 2022)
CDT (Liu et al., 2023b)
TREBI (Lin et al., 2023) |
| Hard Constraint | RL-CBF-CLF-QP (Choi et al., 2021)
ISSA (Zhao et al., 2021)
FAC-SIS (Ma et al., 2022)
RCRL (Yu et al., 2022a)
SRLNBC (Yang et al., 2023a)
SNO-MDP (Wachi & Sui, 2020)
Wang et al. (2023b)
RESPO (Ganai et al., 2024) | FISOR (ours) |

## B   THEORETICAL ANALYSIS

We first analyze the relationship between the feasible-dependent optimization problem in Eq. (4) and the commonly used safe RL optimization objective in Eq. (1). And we provide the proof of key Theorems.

### B.1   FEASIBILITY-DEPENDENT OPTIMIZATION

This section analyzes the relationship between feasibility-dependent optimization and commonly used safe RL optimization objectives from a safety perspective. Therefore, we overlook the KL divergence constraint. The widely used safe RL optimization objective can be written as:

$$\max_{\pi} \; \mathbb{E}_s \left[ V_r^\pi(s) \right] \qquad \text{s.t. } \mathbb{E}_s \left[ V_c^\pi(s) \right] \leq l. \tag{17}$$

When we consider the hard constraint situation, we can set the cost limit to zero and drop the expectation in the constraints:

$$\max_{\pi} \; \mathbb{E}_s \left[ V_r^\pi(s) \right] \qquad \text{s.t. } V_c^\pi(s) \leq 0. \tag{18}$$

First, we divide the state space into two parts: the feasible region and the infeasible region. In the feasible region, where states are denoted as $s \in \mathcal{S}_f$, we can find a policy that satisfies hard constraint in Eq. (18). Conversely, we cannot find a policy that satisfies hard constraints in the infeasible region $s \notin \mathcal{S}_f$. Based on above notations, we introduce Lemma 2 (Proposition 4.2 in Yu et al. (2022a)):

**Lemma 2.** *Eq. (18) is equivalent to:*

$$\max_{\pi} \; \mathbb{E}_s \left[ V_r^\pi(s) \cdot \mathbb{I}_{s \in \mathcal{S}_f} - V_c^\pi(s) \cdot \mathbb{I}_{s \notin \mathcal{S}_f} \right]$$
$$\text{s.t. } V_c^\pi(s) \leq 0, s \in \mathcal{S}_f. \tag{19}$$

For practical implementation, $V_c^\pi(s) \leq 0$ is extremely hard to satisfy due to the approximation error of neural network as shown in the toy case in Figure 1. So we hope to replace $V_c^\pi(s) \leq 0$ with a new safe policy set $\Pi_f(s)$ that is conductive for practical solving. After adding the KL divergence constraint back, we got the feasibility-dependent optimization objective as shown in Eq. (4).

### B.2 PROOF OF LEMMA 1

*Proof.* Due to the involvement of two different optimal policies, one for maximizing rewards ($\max_\pi \mathbb{E}_s [V_r^\pi(s)]$) and the other for minimizing constraint violations ($\max_\pi -V_h^\pi(s)$), in the proof process, in order to avoid confusion, we will use $\pi_r^*$ and $\pi_h^*$ to represent them respectively:

$$
\begin{aligned}
\pi_r^* &\leftarrow \arg\max_\pi \mathbb{E}_s[V_r^\pi(s)], \\
\pi_h^* &\leftarrow \arg\max_\pi -V_h^\pi(s), \forall s.
\end{aligned}
\tag{20}
$$

Next, we will proceed with the proofs of the three subproblems separately.

**Optimization Objective in the Feasible Region**. The expected return of policy $\pi$ in terms of the advantage over another policy $\tilde\pi$ can be expressed as (Schulman et al., 2015; Kakade & Langford, 2002):

$$
\eta(\pi) = \eta(\tilde\pi) + \mathbb{E}_{s\sim d^\pi(s)}\mathbb{E}_{a\sim\pi}\left[A_r^{\tilde\pi}(s,a)\right],
\tag{21}
$$

where $\eta(\pi) := \mathbb{E}_s[V_r^\pi(s)]$ and $d^\pi(s)$ is the discounted visitation frequencies. Considering the case where $\tilde\pi$ is the optimal policy $\pi_r^*$. Eq. (21) can be rewritten as:

$$
\eta(\pi) = \eta(\pi_r^*) + \mathbb{E}_{s\sim d^\pi(s)}\mathbb{E}_{a\sim\pi}\left[A_r^*(s,a)\right],
\tag{22}
$$

According to Eq. (22), we can deduce:

$$
\begin{aligned}
&\max_\pi \ \mathbb{E}_{a\sim\pi}\left[A_r^*(s,a)\right] \\
\overset{\text{i}}{\Rightarrow}\ &\max_\pi \ \mathbb{E}_{s\sim d^\pi(s)}\mathbb{E}_{a\sim\pi}\left[A_r^*(s,a)\right] \\
\overset{\text{ii}}{\Leftrightarrow}\ &\max_\pi \ \eta(\pi_r^*) + \mathbb{E}_{s\sim d^\pi(s)}\mathbb{E}_{a\sim\pi}\left[A_r^*(s,a)\right] \\
\overset{\text{iii}}{\Leftrightarrow}\ &\max_\pi \ \eta(\pi) \\
\overset{\text{iv}}{\Leftrightarrow}\ &\max_\pi \ \mathbb{E}_s\left[V_r^\pi(s)\right].
\end{aligned}
$$

Transformation (i) holds since the optimal policy for $\max_\pi \ \mathbb{E}_{a\sim\pi}\left[A_r^*(s,a)\right]$ guarantees the maximization of expected advantage in any state and thus for the states from the state visitation distribution $d^\pi$. It does not affect the optimality of taking the expected value over states. Transformation (ii) is achieved by adding $\eta(\pi_r^*)$, which is unrelated to the optimization variable $\pi$. Transformation (iii) holds by using Eq. (22). Transformation (iv) holds by using the definition of $\eta(\pi)$.

**Optimization Objective in the Infeasible Region**.

$$
\begin{aligned}
&\max_\pi \ \mathbb{E}_{a\sim\pi}\left[-A_h^*(s,a)\right] \\
\overset{\text{i}}{\Leftrightarrow}\ &\min_\pi \ \mathbb{E}_{a\sim\pi}\left[Q_h^*(s,a)\right] \\
\overset{\text{ii}}{\Leftrightarrow}\ &\min_\pi \ \mathbb{E}_{a\sim\pi}\left[\max_{t\in\mathbb{N}} h(s_t), s_0 = s, a_0 = a, a_{t+1} \sim \pi_h^*(\cdot \mid s_{t+1})\right] \\
\overset{\text{iii}}{\Leftrightarrow}\ &\max_\pi \ -V_h^\pi(s) \\
\Rightarrow\ &\max_\pi \ \mathbb{E}_s\left[-V_h^\pi(s)\right]
\end{aligned}
$$

Transformation (i) is achieved by using the definition of $A_h^*(s,a) := Q_h^*(s,a) - V_h^*(s)$, where $V_h^*(s)$ is unrelated to the action. Transformation (ii) is achieved by using the Definition 1. The definition of $Q_h^*(s,a)$ is that in the first step, action $a$ is taken, and subsequently, the safest policy is followed. Transformation (iii): According to the definition of $\pi_h^*$, $\mathbb{E}_{a\sim\pi}\left[\max_{t\in\mathbb{N}} h(s_t), s_0 = s, a_0 = a, a_{t+1} \sim \pi_h^*(\cdot \mid s_{t+1})\right]$ is minimized if and only if $\pi$ is the optimal policy $\pi_h^*$.

**Safety Constraint in the Feasible Region**.

$$\int_{\{a|Q_h^*(s,a)\leq 0\}} \pi(\cdot|s)\mathrm{d}a = 1$$

$$\overset{\text{i}}{\Leftrightarrow} Q_h^*(s,a)\pi(a|s) \leq 0$$

$$\overset{\text{ii}}{\Rightarrow} V_h^\pi(s) \leq 0$$

Transformation (i): $\int_{\{a|Q_h^*(s,a)\leq 0\}} \pi(\cdot|s)\mathrm{d}a = 1$ means that all possible output action $a$ of the policy need to satisfy $Q_h^*(s,a) \leq 0$. Specifically, for action $a$, if $\pi(a|s)$ is greater than 0, then $Q^*$ must be less than or equal to 0. For actions with a probability density of 0, we no longer require a specific value for $Q_h^*$, meaning that $Q_h^*(s,a)\pi(a|s) \leq 0$. Transformation (ii) can be derived by mathematical induction. In details, $Q_h^*(s,a)\pi(a|s) \leq 0$ implies that for the transition $(s,a) \to s'$, $V^*(s') \leq 0$ holds according to Definition 1. This means the next state $s'$ is also a feasible state. Then, for this feasible $s'$, we can also obtain a feasible state $(s',a') \to s''$ with $V^*(s'') \leq 0$ by constraining $Q_h^*(s',a')\pi(a'|s') \leq 0$. Then, we can derive that all states visited by $\pi$ starting from a feasible region still locate in feasible region, so $Q_h^*(s,a)\pi(a|s) \leq 0 \Rightarrow V_h^\pi(s) \leq 0$ by mathematical inductions.

$\square$

### B.3  PROOF OF THEOREM 1

*Proof.* We first consider the situation when states within feasible region, i.e., $V_h^* \leq 0$. The analytic solution for the constrained optimization problem can be obtained by enforcing the KKT conditions. Then construct the Lagrange function as:

$$L_1(\pi,\lambda_1,\mu_1) = \mathbb{E}_{a\sim\pi}\left[A_r^*(s,a)\right] - \lambda_1\left(\int_{\{a|Q_h^*(s,a)\leq 0\}} \pi(\cdot|s)\mathrm{d}a - 1\right) - \mu_1\left(\mathrm{D}_{\mathrm{KL}}(\pi\|\pi_\beta) - \epsilon\right)$$

Differentiating with respect to $\pi$ gives:

$$\frac{\partial L_1}{\partial \pi} = A_r^*(s,a) - \mu_1 \log \pi_\beta(a|s) + \mu_1 \log \pi(a|s) + \mu_1 - \overline{\lambda}_1$$

where $\overline{\lambda}_1 = \lambda_1 \cdot \mathbb{I}_{Q_h^*(s,a)\leq 0}$. Setting $\frac{\partial L_1}{\partial \pi}$ to zero and solving for $\pi$ gives the closed-form solution:

$$\pi^*(a|s) = \frac{1}{Z_1}\pi_\beta(a|s)\exp\left(\alpha_1 A_r^*(s,a)\right) \cdot \mathbb{I}_{Q_h^*(s,a)\leq 0}$$

where $\alpha_1 = 1/\mu_1$, $Z_1$ is a normalizing constant to make sure that $\pi^*$ is a valid distribution.

For states in the infeasible region, i.e., $V_h^* > 0$, we can use the same method to derivation the closed-form solution. The Lagrange function of the infeasible part is:

$$L_2(\pi,\lambda_2,\mu_2) = \mathbb{E}_{a\sim\pi}\left[-A_h^*(s,a)\right] - \lambda_2\left(\int_a \pi(\cdot|s)\mathrm{d}a - 1\right) - \mu_2\left(\mathrm{D}_{\mathrm{KL}}(\pi\|\pi_\beta) - \epsilon\right)$$

Setting $\frac{\partial L_2}{\partial \pi}$ to zero and solving for $\pi$ gives the closed form solution:

$$\pi^*(a|s) = \frac{1}{Z_2}\pi_\beta(a|s)\exp\left(-\alpha_2 A_h^*(s,a)\right)$$

where $\alpha_2 = 1/\mu_2$, the effect of $Z_2$ is equivalent to that of $Z_1$.

$\square$

## C  GUIDED DIFFUSION POLICY LEARNING WITHOUT TIME-DEPENDENT CLASSIFIER

We first briefly review diffusion models, then present the inherent connections between weighted regression and exact energy guided diffusion sampling (Lu et al., 2023).

## C.1 A RECAP ON DIFFUSION MODEL

Diffusion models (Ho et al., 2020; Song et al., 2021) are powerful generative models that show expressive ability at modeling complex distributions. Following the most notations in (Lu et al., 2023), given an dataset $\mathcal{D} = \{\boldsymbol{x}_0^i\}_{i=1}^D$ with $D$ samples $\boldsymbol{x}_0$ from an unknown data distribution $q_0(\boldsymbol{x}_0)$, diffusion models aim to find an approximation of $q_0(\boldsymbol{x}_0)$ and then generate samples from this approximated distribution.

Diffusion models consist of two processes: forward noising process and reverse denoising process. During the forward nosing process, diffusion models gradually add Gaussian noise to the original data samples $\boldsymbol{x}_0$ from time stamp $0$ to $T > 0$, resulting in the noisy samples $\boldsymbol{x}_T$. The forward noising process from $\boldsymbol{x}_0$ to $\boldsymbol{x}_T$ satisfies the following transition distribution $q_{t0}(\boldsymbol{x}_t|\boldsymbol{x}_0), t \in [0, T]$:

$$q_{t0}(\boldsymbol{x}_t|\boldsymbol{x}_0) = \mathcal{N}(\boldsymbol{x}_t|\alpha_t\boldsymbol{x}_0, \sigma_t^2\boldsymbol{I}), t \in [0, T], \tag{23}$$

where $\alpha_t, \sigma_t > 0$ are fixed noise schedules, which are human-designed to meet the requirements that $q_T(\boldsymbol{x}_T|\boldsymbol{x}_0) \approx q_T(\boldsymbol{x}_T) \approx \mathcal{N}(\boldsymbol{x}_T|0, \tilde{\sigma}^2\boldsymbol{I})$ for some $\tilde{\sigma} > 0$ that is independent of $\boldsymbol{x}_0$, such as Variance Preserving (VP) (Song et al., 2021; Ho et al., 2020) or Variance-Exploding (VE) schedules (Song & Ermon, 2019; Song et al., 2021). Given the forward noising process, one can start from the marginal distribution at $q_T(\boldsymbol{x}_T)$, i.e., $\boldsymbol{x}_T \sim \mathcal{N}(\boldsymbol{x}_T|0, \tilde{\sigma}^2\boldsymbol{I})$ and then reverse the forward process to recover the original data $\boldsymbol{x}_0 \sim q_0(\boldsymbol{x}_0)$. This reverse denoising process can be equivalently solved by solving a diffusion ODE or SDE (Song et al., 2021):

$$\text{(Diffusion ODE)} \quad \mathrm{d}\boldsymbol{x}_t = \left[ f(t)\boldsymbol{x}_t - \frac{1}{2}g^2(t)\nabla_{\boldsymbol{x}_t} \log q_t(\boldsymbol{x}_t) \right] \mathrm{d}t \tag{24}$$

$$\text{(Diffusion SDE)} \quad \mathrm{d}\boldsymbol{x}_t = \left[ f(t)\boldsymbol{x}_t - g^2(t)\nabla_{\boldsymbol{x}_t} \log q_t(\boldsymbol{x}_t) \right] \mathrm{d}t + g(t)\mathrm{d}\bar{\boldsymbol{w}} \tag{25}$$

where $f(t) = \frac{\mathrm{d}\log\alpha_t}{\mathrm{d}t}, g^2(t) = \frac{\mathrm{d}\sigma_t^2}{\mathrm{d}t} - 2\frac{\mathrm{d}\log\alpha_t}{\mathrm{d}t}\sigma_t^2$ are determined by the fixed noise schedules $\alpha_t, \sigma_t$, $\bar{\boldsymbol{w}}$ is a standard Wiener process when time flows backwards from $T$ to $0$. The only unknown element in Eq. (24-25) is the score function $\nabla_{\boldsymbol{x}_t} \log q_t(\boldsymbol{x}_t)$, which can be simply approximated by a neural network $\boldsymbol{z}_\theta(\boldsymbol{x}_t, t)$ via minimizing a MSE loss (Ho et al., 2020; Song et al., 2021):

$$\min_\theta \mathbb{E}_{\boldsymbol{x}_t, t \sim \mathcal{U}([0,T])} \left[ \|\boldsymbol{z}_\theta(\boldsymbol{x}_t, t) + \sigma_t\nabla_{\boldsymbol{x}_t} \log q_t(\boldsymbol{x}_t)\|_2^2 \right]$$
$$\Leftrightarrow \min_\theta \mathbb{E}_{\boldsymbol{x}_t, t \sim \mathcal{U}([0,T]), \boldsymbol{z} \sim \mathcal{N}(\boldsymbol{0}, \boldsymbol{I})} \left[ \|\boldsymbol{z} - \boldsymbol{z}_\theta(\boldsymbol{x}_t, t)\|_2^2 \right] \tag{26}$$

where $\boldsymbol{x}_0 \sim q_0(\boldsymbol{x}_0)$, $\boldsymbol{x}_t = \alpha_t\boldsymbol{x}_0 + \sigma_t\boldsymbol{z}$. Given unlimited model capacity and data, the esitimated $\theta$ satisfies $\boldsymbol{z}_\theta(\boldsymbol{x}_t, t) \approx -\sigma_t\nabla_{\boldsymbol{x}_t} \log q_t(\boldsymbol{x}_t)$. Then, $\boldsymbol{z}_\theta$ can be substituted into Eq. (24-25) and solves the ODEs/SDEs to sample $\boldsymbol{x}_0 \sim q_0(\boldsymbol{x}_0)$.

## C.2 CONNECTIONS BETWEEN WEIGHTED REGRESSION AND EXACT ENERGY GUIDANCE

In our paper, we aim to extract the optimal policy $\pi^*$ in Eq. (12), which behaves as a weighted distribution form as:

$$\pi^*(a|s) \propto \pi_\beta(a|s)w(s, a), \tag{27}$$

where $\pi_\beta$ is the behavior policy that generates the offline dataset $\mathcal{D}$ and $w(s, a)$ is a feasibility-dependent weight function that assigns high values on safe and high-reward $(s, a)$ pairs. This can be abstracted as a standard energy guided sampling problem in diffusion models (Lu et al., 2023):

$$p_0(\boldsymbol{x}_0) \propto q_0(\boldsymbol{x}_0)f(\mathcal{E}(\boldsymbol{x}_0)), \tag{28}$$

where $p_0(\boldsymbol{x}_0)$ is the interested distribution which we want to sample from, $q_0(\boldsymbol{x}_0)$ is parameterized by a pretrained diffusion model trained by Eq. (26), $\mathcal{E}(\boldsymbol{x}_0)$ is any form of energy function that encodes human preferences (e.g. cumulative rewards, cost value functions), $f(x) \geq 0$ can be any non-negative function. To sample $\boldsymbol{x}_0 \sim p_0(\boldsymbol{x}_0)$ with the pretrained $q_0(\boldsymbol{x}_0)$ and the energy function $\mathcal{E}$, previous works typically train a separate time-dependent classifier $\mathcal{E}_t$ and then calculating a modified score function (Lu et al., 2023; Janner et al., 2022; Lin et al., 2023):

$$\nabla_{\boldsymbol{x}_t} \log p_t(\boldsymbol{x}_t) = \nabla_{\boldsymbol{x}_t} \log q_t(\boldsymbol{x}_t) - \nabla_{\boldsymbol{x}_t} f^*(\mathcal{E}_t(\boldsymbol{x}_t)), \tag{29}$$

where $f^*(x)$ is determined by the choice of $f(x)$. Then, one can sample $\boldsymbol{x}_0 \sim p_0(\boldsymbol{x}_0)$ by solving a modified ODEs/SDEs via replacing $\nabla_{\boldsymbol{x}_t} \log q_t(\boldsymbol{x}_t)$ with $\nabla_{\boldsymbol{x}_t} \log p_t(\boldsymbol{x}_t)$ in Eq. (24-25). This kind of guided sampling method requires the training of an additional time-dependent classifier $\mathcal{E}_t$, which is typically hard to train (Lu et al., 2023) and introduces additional training errors.

**Weighted Regression as Exact Energy Guidance.** In this paper, we find that by simply augmenting the energy function into Eq. (26) and forming a weighted regression loss in Eq. (30), we can obtain the $\nabla_{\boldsymbol{x}_t} \log p_t(\boldsymbol{x}_t)$ in Eq. (29) without the training of any time-dependent classifier, which greatly simplifies the training difficulty:

$$\min_{\theta} \mathbb{E}_{\boldsymbol{x}_t, t \sim \mathcal{U}([0,T]), \boldsymbol{z} \sim \mathcal{N}(\mathbf{0}, \boldsymbol{I})} \left[ f(\mathcal{E}(\boldsymbol{x}_0)) \| \boldsymbol{z} - \boldsymbol{z}_\theta(\boldsymbol{x}_t, t) \|_2^2 \right] \tag{30}$$

**Theorem 3.** *We can sample $\boldsymbol{x}_0 \sim p_0(\boldsymbol{x}_0)$ by optimizing the weighted regression loss in Eq. (30) and substituting the obtained $\boldsymbol{z}_\theta$ into the diffusion ODEs/SDEs in Eq. (24-25).*

*Proof.* We denote $q_t(\boldsymbol{x}_t)$ and $p_t(\boldsymbol{x}_t)$ as the marginal distribution of the forward noising process at time $t$ starting from $q_0(\boldsymbol{x}_0)$ and $p_0(\boldsymbol{x}_0)$, respectively. We let the forward noising processes of $q_{t0}(\boldsymbol{x}_t|\boldsymbol{x}_0)$ and $p_{t0}(\boldsymbol{x}_t|\boldsymbol{x}_0)$ share the same transition distribution, that is

$$q_{t0}(\boldsymbol{x}_t|\boldsymbol{x}_0) = p_{t0}(\boldsymbol{x}_t|\boldsymbol{x}_0) = \mathcal{N}(\boldsymbol{x}_t|\alpha_t \boldsymbol{x}_0, \sigma_t^2 \boldsymbol{I}), t \in [0, T], \tag{31}$$

Then, we have

$$\begin{aligned} p_t(\boldsymbol{x}_t) &= \int p_{t0}(\boldsymbol{x}_t|\boldsymbol{x}_0) p_0(\boldsymbol{x}_0) \mathrm{d}\boldsymbol{x}_0 \\ &= \int q_{t0}(\boldsymbol{x}_t|\boldsymbol{x}_0) \frac{q_0(\boldsymbol{x}_0) f(\mathcal{E}(\boldsymbol{x}_0))}{Z} \mathrm{d}\boldsymbol{x}_0 \end{aligned} \tag{32}$$

where the first equation holds for the definition of marginal distribution, the second equation holds for the definition of $p_0(\boldsymbol{x}_0) \propto q_0(\boldsymbol{x}_0) f(\mathcal{E}(\boldsymbol{x}_0))$ and $Z = \int q_0(\boldsymbol{x}_0) f(\mathcal{E}(\boldsymbol{x}_0)) \mathrm{d}\boldsymbol{x}_0$ is a normalization constant to make $p_0$ a valid distribution.

Then, taking the derivation w.r.t $\boldsymbol{x}_t$ on the logarithm of each side of Eq. (32), we have

$$\begin{aligned} \nabla_{\boldsymbol{x}_t} \log p_t(\boldsymbol{x}_t) &= \nabla_{\boldsymbol{x}_t} \log \int q_{t0}(\boldsymbol{x}_t|\boldsymbol{x}_0) q_0(\boldsymbol{x}_0) f(\mathcal{E}(\boldsymbol{x}_0)) \mathrm{d}\boldsymbol{x}_0 \\ &= \frac{\int \nabla_{\boldsymbol{x}_t} q_{t0}(\boldsymbol{x}_t|\boldsymbol{x}_0) q_0(\boldsymbol{x}_0) f(\mathcal{E}(\boldsymbol{x}_0)) \mathrm{d}\boldsymbol{x}_0}{\int q_{t0}(\boldsymbol{x}_t|\boldsymbol{x}_0) q_0(\boldsymbol{x}_0) f(\mathcal{E}(\boldsymbol{x}_0)) \mathrm{d}\boldsymbol{x}_0} \\ &= \frac{\int \nabla_{\boldsymbol{x}_t} q_{t0}(\boldsymbol{x}_t|\boldsymbol{x}_0) q_0(\boldsymbol{x}_0) f(\mathcal{E}(\boldsymbol{x}_0)) \mathrm{d}\boldsymbol{x}_0}{p_t(\boldsymbol{x}_t)} \\ &= \frac{\int q_{t0}(\boldsymbol{x}_t|\boldsymbol{x}_0) q_0(\boldsymbol{x}_0) f(\mathcal{E}(\boldsymbol{x}_0)) \nabla_{\boldsymbol{x}_t} \log q_{t0}(\boldsymbol{x}_t|\boldsymbol{x}_0) \mathrm{d}\boldsymbol{x}_0}{p_t(\boldsymbol{x}_t)}, \end{aligned} \tag{33}$$

where the first equation holds since $Z$ is a constant w.r.t $\boldsymbol{x}_t$ and thus can be dropped. To approximate the score function $\nabla_{\boldsymbol{x}_t} \log p_t(\boldsymbol{x}_t)$, we can resort to denoising score matching (Song & Ermon, 2019) via optimizing the following objective:

$$\min_\theta \int p_t(\boldsymbol{x}_t) \left[ \|\boldsymbol{z}_\theta(\boldsymbol{x}_t, t) + \sigma_t \nabla_{\boldsymbol{x}_t} \log p_t(\boldsymbol{x}_t)\|_2^2 \right] \mathrm{d}\boldsymbol{x}_t$$

$$\overset{i}{\Leftrightarrow} \min_\theta \int p_t(\boldsymbol{x}_t) \left[ \left\| \boldsymbol{z}_\theta(\boldsymbol{x}_t, t) + \sigma_t \frac{\int q_{t0}(\boldsymbol{x}_t|\boldsymbol{x}_0) q_0(\boldsymbol{x}_0) f(\mathcal{E}(\boldsymbol{x}_0)) \nabla_{\boldsymbol{x}_t} \log q_{t0}(\boldsymbol{x}_t|\boldsymbol{x}_0) \mathrm{d}\boldsymbol{x}_0}{p_t(\boldsymbol{x}_t)} \right\|_2^2 \right] \mathrm{d}\boldsymbol{x}_t$$

$$\overset{ii}{\Leftrightarrow} \min_\theta \int p_t(\boldsymbol{x}_t) \left[ \|\boldsymbol{z}_\theta(\boldsymbol{x}_t, t)\|_2^2 + 2\sigma_t \boldsymbol{z}_\theta(\boldsymbol{x}_t, t) \frac{\int q_{t0}(\boldsymbol{x}_t|\boldsymbol{x}_0) q_0(\boldsymbol{x}_0) f(\mathcal{E}(\boldsymbol{x}_0)) \nabla_{\boldsymbol{x}_t} \log q_{t0}(\boldsymbol{x}_t|\boldsymbol{x}_0) \mathrm{d}\boldsymbol{x}_0}{p_t(\boldsymbol{x}_t)} \right] \mathrm{d}\boldsymbol{x}_t$$

$$\overset{iii}{\Leftrightarrow} \min_\theta \int \int \int q_0(\boldsymbol{x}_0) q_{t0}(\boldsymbol{x}_t|\boldsymbol{x}_0) f(\mathcal{E}(\boldsymbol{x}_0)) \left[ \|\boldsymbol{z}_\theta(\boldsymbol{x}_t, t) + \sigma_t \nabla_{\boldsymbol{x}_t} \log q_{t0}(\boldsymbol{x}_t|\boldsymbol{x}_0)\|_2^2 \right] \mathrm{d}\boldsymbol{x}_0 \mathrm{d}t \mathrm{d}\boldsymbol{z}$$

$$\overset{iv}{\Leftrightarrow} \min_\theta \mathbb{E}_{\boldsymbol{x}_0 \sim q_0(\boldsymbol{x}_0), t \sim \mathcal{U}([0,T]), \boldsymbol{z} \sim \mathcal{N}(\boldsymbol{0}, \boldsymbol{I})} \left[ f(\mathcal{E}(\boldsymbol{x}_0)) \|\boldsymbol{z}_\theta(\boldsymbol{x}_t, t) - \boldsymbol{z}\|_2^2 \right]$$

$$(34)$$

where $\boldsymbol{x}_t = \alpha_t \boldsymbol{x}_0 + \sigma_t \boldsymbol{z}$. Transformation (i) holds by substituting Eq. (33) into $\nabla_{\boldsymbol{x}_t} \log p_t(\boldsymbol{x}_t)$. Transformation (ii) holds by expanding the L2 norm according to its definition and drop the terms that are independent to $\theta$. Transformation (iii) holds by rearranging (ii). Transformation (iv) holds since $q_{t0}(\boldsymbol{x}_t|\boldsymbol{x}_0) = \mathcal{N}(\boldsymbol{x}_t|\alpha_t \boldsymbol{x}_0, \sigma_t^2 \boldsymbol{I})$, which is re-parameterized as $\boldsymbol{x}_t = \alpha_t \boldsymbol{x}_0 + \sigma_t \boldsymbol{z}, \boldsymbol{z} \sim \mathcal{N}(\boldsymbol{0}, \boldsymbol{I})$. Therefore, $\sigma_t \nabla_{\boldsymbol{x}_t} \log q_{t0}(\boldsymbol{x}_t|\boldsymbol{x}_0) = -\boldsymbol{z}$. Eq. (34) is exactly the weighed regression loss in Eq. (30), meaning that the exact $p_0(\boldsymbol{x}_0)$ can be obtained by training this simple weighted regression loss and completes the proof. □

Theorem 2 can be easily obtained via replacing $p_0$ with $\pi^*$, replacing $q_0$ with $\pi_\beta$ and replacing $f(\mathcal{E})$ with $w$. Some recent works also mention the weighted regression form (Hansen-Estruch et al., 2023; Kang et al., 2023), but they do not identify these inherent connections between weighted regression and exact energy guidance. Based on Theorem 3, we show that we can extract $\pi^*$ using the weighted regression loss in Eq. (14).

### C.3 COMPARISONS WITH OTHER GUIDED SAMPLING METHODS

Diffusion models are powerful generative models. The guided sampling methods in diffusion models allow humans to embed their preferences to guide the diffusion models to generate desired outputs (Lu et al., 2023; Ajay et al., 2023; Janner et al., 2022; Ho & Salimans, 2021; Dhariwal & Nichol, 2021), such as the policy that obtains the highest rewards and adheres to zero constraint violations.

**Classifier-Free Guidance**. One of the most popular guided sampling methods is the classifier-free guided sampling method (Ajay et al., 2023; Ho & Salimans, 2021), which trains the diffusion model with the condition information assistance to implicitly achieve classifier guidance. However, the conditioned variables may not be always easy to acquire. In contrast, it may be easy for us to obtain a scalar function that encodes our preferences such as the cost value or reward value functions.

**Classifier Guided Sampling**. The other most popular guided sampling method is classifier guided sampling (Lu et al., 2023; Janner et al., 2022; Dhariwal & Nichol, 2021), which trains a separate time-dependent classifier and uses its gradient to modify the score function during sampling like Eq. (29) does. This guided sampling method requires the training of an additional time-dependent classifier, which is typically hard to train and may suffer from large approximation errors (Lu et al., 2023). In contrast, we smartly encode the classifier information into the diffusion model in the training stage through the weighted regression loss in Eq. (30), which avoids the training of the complicated time-dependent classifier, greatly simplifying the training.

**Direct Guided Sampling**. Some diffusion-based offline RL methods directly use diffusion models to parameterize the policy and update diffusion models to directly maximize the Q value (Wang et al., 2023c). This requires the gradient backward through the entire reverse denoising process, inevitably introducing tremendous computational costs.

**Sample-based Guided Sampling**. Some diffusion-based offline RL methods directly use diffusion models to fit the behavior policy $\pi_\beta$ and then sample a lot of action candidates from the approximated $\pi_\beta$ and then select the best one based on the classifier signal such as value functions (Hansen-Estruch et al., 2023; Chen et al., 2023). Although its simplicity, these methods show surprisingly good

performances. Inspired by this, we also generate some action candidates and select the safest one to enhance safety.

# D    EXPERIMENTAL DETAILS

This section outlines the experimental details to reproduce the main results in our papers.

## D.1    DETAILS ON TOY CASE EXPERIMENTS

The toy case experiments in this paper primarily focus on the reach-avoid control task (Yang et al., 2023a; Ma et al., 2021a), where the agent is required to navigate to the target location while avoiding hazards. The state space of the agent can be represented as: $\mathcal{S} := (x, y, v, \theta)$, where $x$ and $y$ correspond to the agent's coordinates, $v$ represents velocity, and $\theta$ represents the yaw angle. The action space of the agent can be represented as: $\mathcal{A} := (\dot{v}, \dot{\theta})$, where $\dot{v}$ corresponds to the acceleration, $\dot{\theta}$ represents angular acceleration. The reward function $r$ is defined as the difference between the distance to the target position at the previous time step and the distance at the current time step. A larger change in distance implies a higher velocity and results in a higher reward. The constraint violation function $h$ is defined as:

$$h := R_{\text{hazard}} - \min\{d_{\text{hazard1}}, d_{\text{hazard2}}\}$$

where $R_{\text{hazard}}$ is the radius of the hazard. $d_{\text{hazard1}}, d_{\text{hazard2}}$ are the distances between the agent and two hazards. When $h \leq 0$, the agent collides with the hazard, otherwise the agent is in a safe state. Correspondingly, $c$ is defined as $c := \max(h, 0)$.

To obtain the ground truth largest feasible region, based on the current agent's state, we use the safest policy (maximum deceleration, maximum angular acceleration away from hazards). If no collision occurs, the current state is within the largest feasible region.

We use the converged DRPO (Yu et al., 2022b) algorithm to collect 50k of interactive data, and use a randomly initialized policy to collect 50k of data, totaling 100k, for FISOR training. The data distribution is illustrated in Figure 1 (b).

In the toy case environment, we also conduct ablation experiments focusing on the selection of parameter $\tau$ in Eq. (8) and visualized the feasible region in Figure 6. See from the results that with a small $\tau$ value, the learned feasible region will become smaller and more conservative. This is acceptable since a smaller feasible region will not likely induce false negative infeasible regions and is beneficial to induce safe policies.

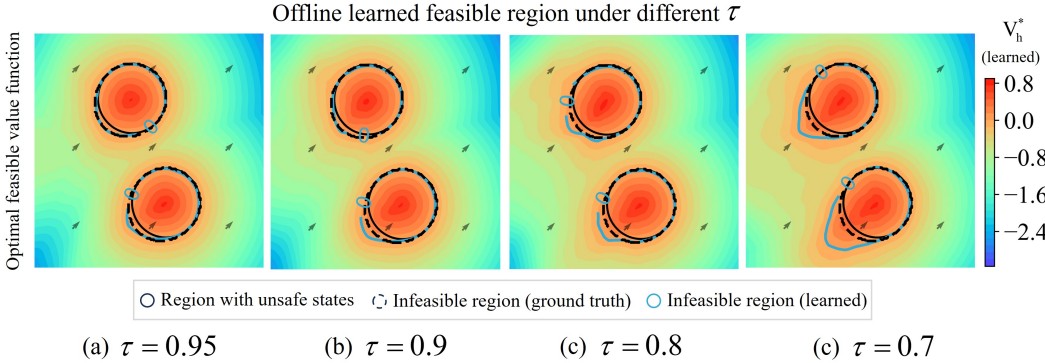

(a) $\tau = 0.95$      (b) $\tau = 0.9$      (c) $\tau = 0.8$      (c) $\tau = 0.7$

Figure 6: Visualization of feasible region under different $\tau$.

## D.2    TASK DESCRIPTION

**Safety-Gymnasium** (Ray et al., 2019; Ji et al., 2023). Environments based on the Mujoco physics simulator. For the agent Car there are three tasks: Button, Push, Goal. 1 and 2 are used to

represent task difficulty. In these tasks, agents need to reach the goal while avoiding hazards. The tasks are named as {Agent}{Task}{Difficulty}. Safety-Gymnasium also provides three velocity constraint tasks based on Ant, HalfCheetah, and Swimmer. Figure 7 visualizes the tasks in Safety-Gymnasium.

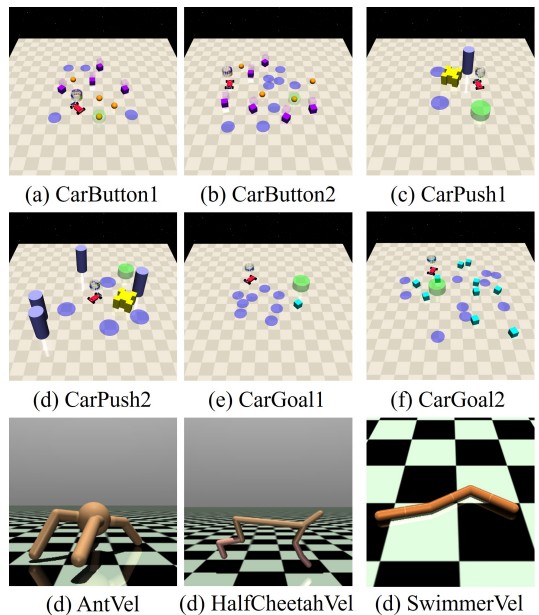

(a) CarButton1     (b) CarButton2     (c) CarPush1

(d) CarPush2     (e) CarGoal1     (f) CarGoal2

(d) AntVel     (d) HalfCheetahVel     (d) SwimmerVel

Figure 7: Visualization of the Safety-Gymnasium environments.

**Bullet-Safety-Gym** (Gronauer, 2022). Environments based on the PyBullet physics simulator. There are four types of agents: Ball, Car, Drone, Ant, and two types of tasks: Circle, Run. The tasks are named as {Agent}{Task}. Figure 8 (a) visualizes the tasks in Bullet-Safety-Gym.

**MetaDrive** (Li et al., 2022). Environments based on the Panda3D game engine simulate real-world driving scenarios. The tasks are named as {Road}{Vehicle}. Road includes three different levels for self-driving cars: easy, medium, hard. Vehicle includes four different levels of surrounding traffic: sparse, mean, dense. Figure 8 (b) visualizes the tasks in MetaDrive.

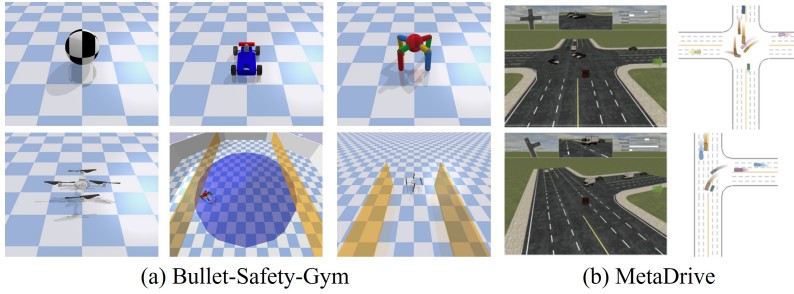

(a) Bullet-Safety-Gym        (b) MetaDrive

Figure 8: Visualization of the Bullet-Safety-Gym and MetaDrive environments.

### D.3 ADDITIONAL EXPERIMENTAL DETAILS

**Evaluation metrics**. Our evaluation protocol strictly follows the recent safe offline RL benchmark DSRL (Liu et al., 2023a), we use the normalized reward return and the normalized cost returns as

https://www.safety-gymnasium.com/en/latest/
https://github.com/liuzuxin/Bullet-Safety-Gym/
https://github.com/liuzuxin/DSRL

the evaluation metrics: $R_{\text{normalized}} = \frac{R_\pi - r_{\min}}{r_{\max} - r_{\min}}$, $C_{\text{normalized}} = \frac{C_\pi + \epsilon}{l + \epsilon}$, where $R_\pi = \sum_t r_t$ is the policy's reward return and $C_\pi = \sum_t c_t$ is its cost return. In all the benchmark tasks, the per-step cost $c_t = 0$ if the state is safe and $c_t > 0$ for unsafe states (Ray et al., 2019; Gronauer, 2022; Li et al., 2022). $r_{\min}$ and $r_{\max}$ are the minimum and maximum reward return in the offline data, respectively. $l$ is a human-defined cost limit. $\epsilon$ is a positive number to ensure numerical stability if $l = 0$. Following the DSRL benchmark, the task is considered safe when $C_\pi$ does not exceed the cost limit $l$, or in other words $C_{\text{normalized}}$ is no larger than 1.

**Feasible Value Function Learning**. According to the loss function of the feasible value functions in Eq. (9), we need to perform function learning using the constraint violation function $h(s)$. However, the DSRL (Liu et al., 2023a) benchmark used in the experiment does not provide it, but only provides a cost function $c(s)$. Therefore, we simply employ a sparse $h(s)$ function design. Specifically, when $c(s) = 0$, the state is safe, $h(s) = -1$; when $c > 0$, the state is unsafe, $h = M(M > 0)$. For optimal algorithm performance, we observe that it works best when the predicted mean of the feasible value function is around 0. Based on this, we consistently chose $M = 25$ across all experiments. Note that determining the value of $M$ doesn't need online evaluations for hyperparameter selection since we only need to monitor the mean value of $V_h^*$ without any online interactions.

**Network Architecture and Training Details**. We implement the feasible value functions and value functions with 2-layer MLPs with ReLU activation functions and 256 hidden units for all networks. During the training, we set the $\tau$ for expectile regression in Eq. (8) and Eq. (15) to 0.9. And we use clipped double Q-learning (Fujimoto et al., 2018), taking a minimum of two $Q_r$ and a maximum of two $Q_h$. We update the target network with $\alpha = 0.001$. Following Kostrikov et al. (2022), we clip exponential advantages to $(-\infty, 100]$ in feasible part and $(-\infty, 150]$ in infeasible part. And we set the temperatures in Eq. (13) with $\alpha_1 = 3$ and $\alpha_2 = 5$. In our paper, we use the diffusion model in IDQL (Hansen-Estruch et al., 2023), which is implemented by JAXRL.

### D.4 PSEUDOCODE AND COMPUTATIONAL COST

The pseudocode of FISOR is presented in Algorithm 1. We implement our approach using the JAX framework (Bradbury et al., 2018). On a single RTX 3090 GPU, we can perform 1 million gradient steps in approximately 45 minutes for all tasks.

---

**Algorithm 1** Feasibility-Guided Safe Offline RL (FISOR)

---

Initialize networks $Q_h, V_h, Q_r, V_c, z_\theta$.
Optimal feasible value function learning:
**for** each gradient step **do**
    Update $V_h$ using Eq. (8)
    Update $Q_h$ using Eq. (9)
**end for**
Optimal value function learning (IQL):
**for** each gradient step **do**
    Update $V_r$ using Eq. (15)
    Update $Q_r$ using Eq. (16)
**end for**
Guided diffusion policy learning:
**for** each gradient step **do**
    Update $z_\theta$ using Eq. (14)
**end for**

---

### D.5 HYPERPARAMETERS

We use Adam optimizer with a learning rate $3e^{-4}$ for all networks. The batch size is set to 256 for value networks and 2048 for the diffusion model. We report the detailed setup in Table 5.

---

https://github.com/ikostrikov/jaxrl

Table 5: Hyperparameters of FISOR

| Parameter | Value |
|-----------|-------|
| Optimizer | Adam |
| Learning rate | 3e-4 |
| Value function batch size | 256 |
| Diffusion model batch size | 2048 |
| Number of hidden layers (value function) | 2 |
| Number of neurons in a hidden layer | 256 |
| Activation function | ReLU |
| Expectile $\tau$ | 0.9 |
| Feasible temperature $\alpha_1$ | 3 |
| Infeasible temperature $\alpha_2$ | 5 |
| Discount factor $\gamma$ | 0.99 |
| Soft update $\alpha$ | 0.001 |
| Exponential advantages clip (feasible) | $(-\infty, 100]$ |
| Exponential advantages clip (infeasible) | $(-\infty, 150]$ |
| Number of times Gaussian noise is added $T$ | 5 |
| Number of action candidates $N$ | 16 |
| Training steps | 1e6 |

### D.6 DETAILS ON SAFE OFFLINE IMITATION LEARNING

We can also extend FISOR to the domain of safe offline imitation learning (IL), showcasing the strong adaptability of the algorithm.

**Task Description**. We test FISOR in three MetaDrive (Li et al., 2022) environments: `mediumsparse`, `mediummean`, `mediumdense`. Based on the dataset provided by DSRL (Liu et al., 2023a), we select trajectories with high rewards that are larger than 220 (expert dataset), including both safe and unsafe trajectories, as shown in Figure 9. The dataset does not include reward information but the cost function is retained.

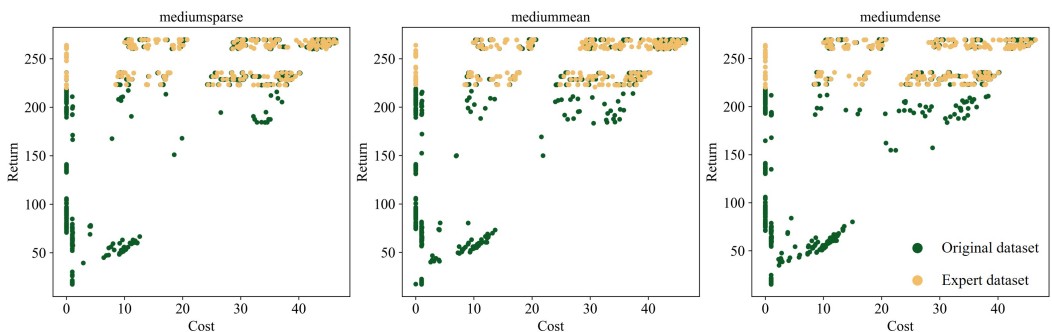

Figure 9: Visualization of the expert dataset trajectories on the cost-return space.

**Baselines**. To apply FISOR in the safe offline IL setting, it suffices to modify Eq. (13) to:

$$w_{IL}(s,a) = \begin{cases} \mathbb{I}_{Q_h^*(s,a) \leq 0} & V_h^*(s) \leq 0 \text{ (Feasible)} \\ \exp\left(-\alpha_2 A_h^*(s,a)\right) & V_h^*(s) > 0 \text{ (Infeasible)} \end{cases}$$

In the IL experiment, we adjust the batch size of the diffusion model to 1024, set the constraint Violation scale $M$ to 1, and keep the rest consistent with Table 5.

BC-P adds a fixed safety penalty on top of BC (akin to TD3+BC (Fujimoto & Gu, 2021)), and its loss can be written as:

$$\text{BC-P}: \quad \min_\theta \mathcal{L}_\pi(\theta) = \min_\theta \mathbb{E}_{(s,a)\sim\mathcal{D}} \left[ (\pi_\theta(s) - a)^2 + k \cdot Q_c^\pi(s,a) \right],$$

where $k$ is a constant used to control the level of punishment and is set as $k = 5$.

Compared to BC-P, BC-L replaces $k$ with an adaptive PID-based Lagrangian multiplier (Stooke et al., 2020), and introduces a cost limit $l = 10$. Its loss can be written as:

$$\text{BC-L}: \quad \min_\theta \mathcal{L}_\pi(\theta) = \min_\theta \mathbb{E}_{(s,a)\sim\mathcal{D}} \left[ (\pi_\theta(s) - a)^2 + \lambda \cdot (Q_c^\pi(s,a) - l) \right].$$

Each experiment result is averaged over 20 evaluation episodes and 5 seeds.

### D.7 EXPERIMENTS OF EXTENDING SAFE ONLINE RL TO OFFLINE SETTINGS

Besides the safe offline RL baselines, we adapted some safe online RL methods to offline settings. Specifically, we consider Sauté (Sootla et al., 2022) and RCRL (Yu et al., 2022a), as Sauté has good versatility and RCRL also utilizes HJ Reachability to enforce hard constraints.

However, directly applying safe online RL methods in offline settings faces the distributional shift challenge. Moreover, it is also difficult to strike the right balance among three highly intricate and correlated aspects: safety constraint satisfaction, reward maximization, and behavior regularization in offline settings. Considering these challenges, we added a behavior regularization term like TD3+BC (Fujimoto & Gu, 2021) into Sauté and RCRL to combat the distributional shift. Also, we carefully tuned the conservatism strength to find a good balance that can obtain good results.

In particular, Sauté (Sootla et al., 2022) handles the safety constraints by integrating them into the state space and modifying the objective accordingly, which can be easily applied to both online RL methods. We incorporated this method into the state-of-the-art offline RL algorithm, TD3+BC (Fujimoto & Gu, 2021) (*Sauté-TD3BC*). The policy loss can be written as:

$$\text{Sauté-TD3BC}: \quad \min_\theta \mathcal{L}_\pi(\theta) = \min_\theta \mathbb{E}_{(s,a)\sim\mathcal{D}} \left[ (\pi_\theta(s) - a)^2 - \lambda \cdot \acute{Q}_r^\pi(s,a) \right],$$

where $\acute{Q}_r^\pi$ is the Sautéd action-value function.

RCRL (Yu et al., 2022a), based on reachability theory (hard constraint) and using Lagrangian iterative solving, can be combined with TD3+BC to mitigate potential distributional shift issues in offline settings (*RC-TD3BC*). Its loss can be written as:

$$\text{RC-TD3BC}: \quad \min_\theta \mathcal{L}_\pi(\theta) = \min_\theta \mathbb{E}_{(s,a)\sim\mathcal{D}} \left[ (\pi_\theta(s) - a)^2 - \lambda_1 \cdot Q_r^\pi(s,a) + \lambda_2(s) Q_h^\pi(s,a) \right].$$

We test the Sauté-TD3BC and RC-TD3BC on DSRL datasets and carefully tune the hyper-parameters to find good results, the results are presented in Table 6. However, these methods fail to obtain good results even with carefully swept hyper-parameters due to the intricate coupling objectives including reward maximization, safety satisfaction, and distributional shift. In detail, Sauté-TD3BC still permits some constraint violations, failing to effectively balance maximizing rewards and meeting safety constraints. RC-TD3BC uses hard constraints, but its coupled solving approach tends to make the algorithm overly conservative. Despite adopting the TD3+BC solving method, RC-TD3BC still faces distributional shift issues. Therefore, naively extending safe online RL algorithms directly to offline settings can lead to reduced algorithm performance and issues like distributional shifts.

### D.8 DATA QUANTITY SENSITIVITY EXPERIMENTS.

We select some competitive baselines that achieve relatively good safety and reward performances in Table 1 and train them with $1/2$ and $1/10$ of the data volume. Figure 10 shows that all baselines fail miserably under the small data setting that only contains $1/10$ of the original data. FISOR,

Table 6: Results of extending safe online RL to offline settings. ↑ means the higher the better. ↓ means the lower the better. Each value is averaged over 20 evaluation episodes and 3 seeds. Gray: Unsafe agents. **Bold**: Safe agents whose normalized cost is smaller than 1. : Safe agent with the highest reward.

| Task | Sauté-TD3BC | | RC-TD3BC | | FISOR (ours) | |
|------|-------------|--------|----------|--------|--------------|--------|
| | reward ↑ | cost ↓ | reward ↑ | cost ↓ | reward ↑ | cost ↓ |
| CarButton1 | -0.03 | 3.48 | -0.10 | 3.89 | **-0.02** | **0.26** |
| CarButton2 | -0.06 | 5.48 | -0.12 | 2.30 | **0.01** | **0.58** |
| CarPush1 | **0.19** | **0.97** | 0.24 | 1.81 | **0.28** | **0.28** |
| CarPush2 | 0.05 | 3.03 | 0.16 | 3.48 | **0.14** | **0.89** |
| CarGoal1 | 0.34 | 1.56 | 0.44 | 3.84 | **0.49** | **0.83** |
| CarGoal2 | 0.23 | 3.36 | 0.33 | 4.18 | **0.06** | **0.33** |
| AntVel | 0.80 | 1.12 | **-1.01** | **0.00** | **0.89** | **0.00** |
| HalfCheetahVel | 0.95 | 11.87 | **-0.37** | **0.21** | **0.89** | **0.00** |
| SwimmerVel | 0.44 | 8.95 | -0.03 | 1.09 | **-0.04** | **0.00** |
| **SafetyGym Average** | 0.32 | 4.42 | -0.05 | 2.31 | **0.30** | **0.35** |
| AntRun | 0.48 | 1.56 | **0.02** | **0.00** | **0.45** | **0.03** |
| BallRun | 0.24 | 3.43 | -0.08 | 16.55 | **0.18** | **0.00** |
| CarRun | 0.87 | 1.50 | -0.24 | 33.35 | **0.73** | **0.14** |
| DroneRun | 0.42 | 13.38 | -0.19 | 9.8 | **0.30** | **0.55** |
| AntCircle | 0.22 | 17.57 | **0.00** | **0.00** | **0.20** | **0.00** |
| BallCircle | 0.61 | 1.13 | 0.10 | 29.18 | **0.34** | **0.00** |
| CarCircle | 0.61 | 3.81 | 0.01 | 45.11 | **0.40** | **0.11** |
| DroneCircle | -0.11 | 1.22 | -0.25 | 3.04 | **0.48** | **0.00** |
| **BulletGym Average** | 0.42 | 5.45 | -0.08 | 17.13 | **0.39** | **0.10** |
| easysparse | 0.01 | 6.19 | **-0.06** | **0.24** | **0.38** | **0.53** |
| easymean | 0.31 | 3.61 | **-0.06** | **0.20** | **0.38** | **0.25** |
| easydense | **-0.03** | **0.49** | **-0.05** | **0.12** | **0.36** | **0.25** |
| mediumsparse | 0.37 | 2.25 | **-0.08** | **0.33** | **0.42** | **0.22** |
| mediummean | 0.01 | 6.19 | **-0.07** | **0.10** | **0.39** | **0.08** |
| mediumdense | 0.11 | 1.62 | **-0.07** | **0.20** | **0.49** | **0.44** |
| hardsparse | 0.26 | 4.37 | **-0.04** | **0.22** | **0.30** | **0.01** |
| hardmean | 0.11 | 2.00 | **-0.03** | **0.56** | **0.26** | **0.09** |
| harddense | **-0.02** | **0.42** | **-0.03** | **0.48** | **0.30** | **0.34** |
| **MetaDrive Average** | 0.13 | 3.02 | **-0.05** | **0.27** | **0.36** | **0.25** |

however, still meets safety requirements and demonstrates more stable performance compared to other methods, although a reduction in data volume weakens FISOR's safety a little. We believe FISOR enjoys such good stability as it decouples the intricate training processes of safe offline RL, which greatly enhances training performances.

# E LIMITATIONS AND DISCUSSIONS

A limitation of FISOR is its requirement for more hyper-parameter tuning, such as the forward/reverse expectile and policy extraction temperatures. However, our experiments demonstrate that the algorithm's performance is not sensitive to hyper-parameter variations. Using only a single set of hyper-parameters, FISOR consistently outperforms baselines and obtains great safety/reward performances on 26 tasks. Moreover, safety constraints with disturbances or probabilistic constraint can be challenging for FISOR. These constraints may affect network estimations and impact the algorithm's performance.

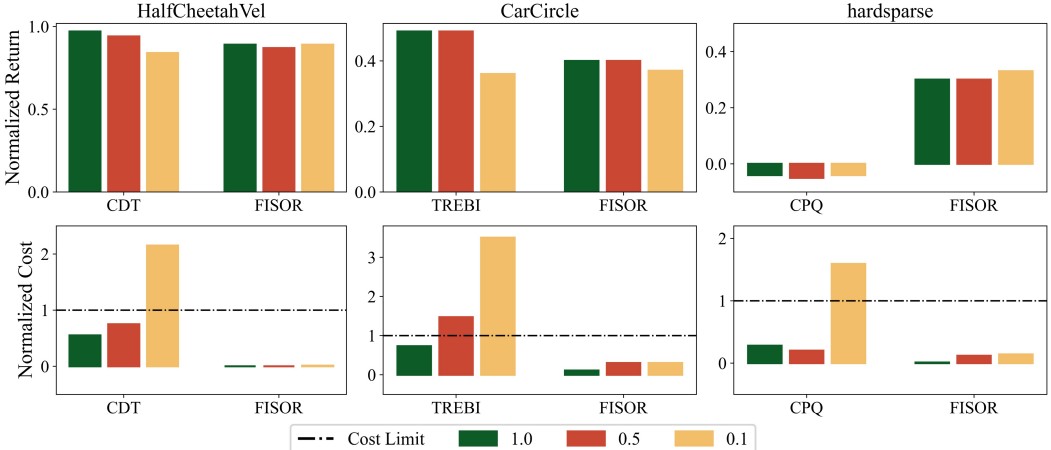

Figure 10: Data quantity sensitivity experiment results

While FISOR performs well on most tasks, it still struggles to achieve zero constraint violations in some cases. We attempt to analyze this issue and identify the following potential reasons:

- We use $Q_h^*$ to construct safety constraint, which in theory can transform to $V_h^\pi$ as stated in Appendix B.2. However, in the offline setting, we can only conduct constraints on the state distribution $d^{\mathcal{D}}$ induced by the offline dataset $\mathcal{D}$ rather than the whole state space. Therefore, the transfer from $V_h^\pi$ to $Q_h^*$ requires another assumption on the data coverage that $0 \le \frac{d^\pi(s)}{d^{\mathcal{D}}(s)} \le C$ since $Q_h^*(s,a)d^{\mathcal{D}}(s)\pi(a|s) \Rightarrow Q_h^*(s,a)d^\pi(s)\pi(a|s)$ only when the offline data has a good coverage on the policy distribution $d^\pi$. Strictly enforcing the policy distribution $d^\pi$ staying within the support of offline data distribution, however, remains a longstanding challenge for offline RL (Lee et al., 2021; Levine et al., 2020). One can use $V_h^\pi$ to enforce safety constraints. However, training the $V_h^\pi$ is coupled with its policy training, which hinders algorithm stability as discussed before.

- In the offline setting, it is difficult to obtain the true value of $V_h^*$ and $V_r^*$. Instead, we only have access to a near-optimal value function within the data distribution. Learning the optimal solution with limited data also remains a challenging problem (Kostrikov et al., 2022; Xu et al., 2023; Garg et al., 2023; Xiao et al., 2023).

Despite the inherent challenges of offline policy learning, to the best of our knowledge, our work represents a pioneering effort in considering hard constraints within the offline setting. Moreover, we disentangle and separately address the complex objectives of reward maximization, safety constraint adherence, and behavior regularization for safe offline RL. This approach involves training these objectives independently, which enhances training stability and makes it particularly suitable for offline scenarios. In our empirical evaluations, we demonstrate that, compared to soft constraints, our approach consistently achieves the lowest constraint violations without the need for selecting a cost limit and attains the highest returns across most tasks, showcasing the promise of applying hard constraints and decoupled training in safe offline RL.

## F  LEARNING CURVES

We test FISOR with the same set of parameters in Table 5 in 26 tasks of DSRL (Liu et al., 2023a), and the learning curve is shown in Figure 11.

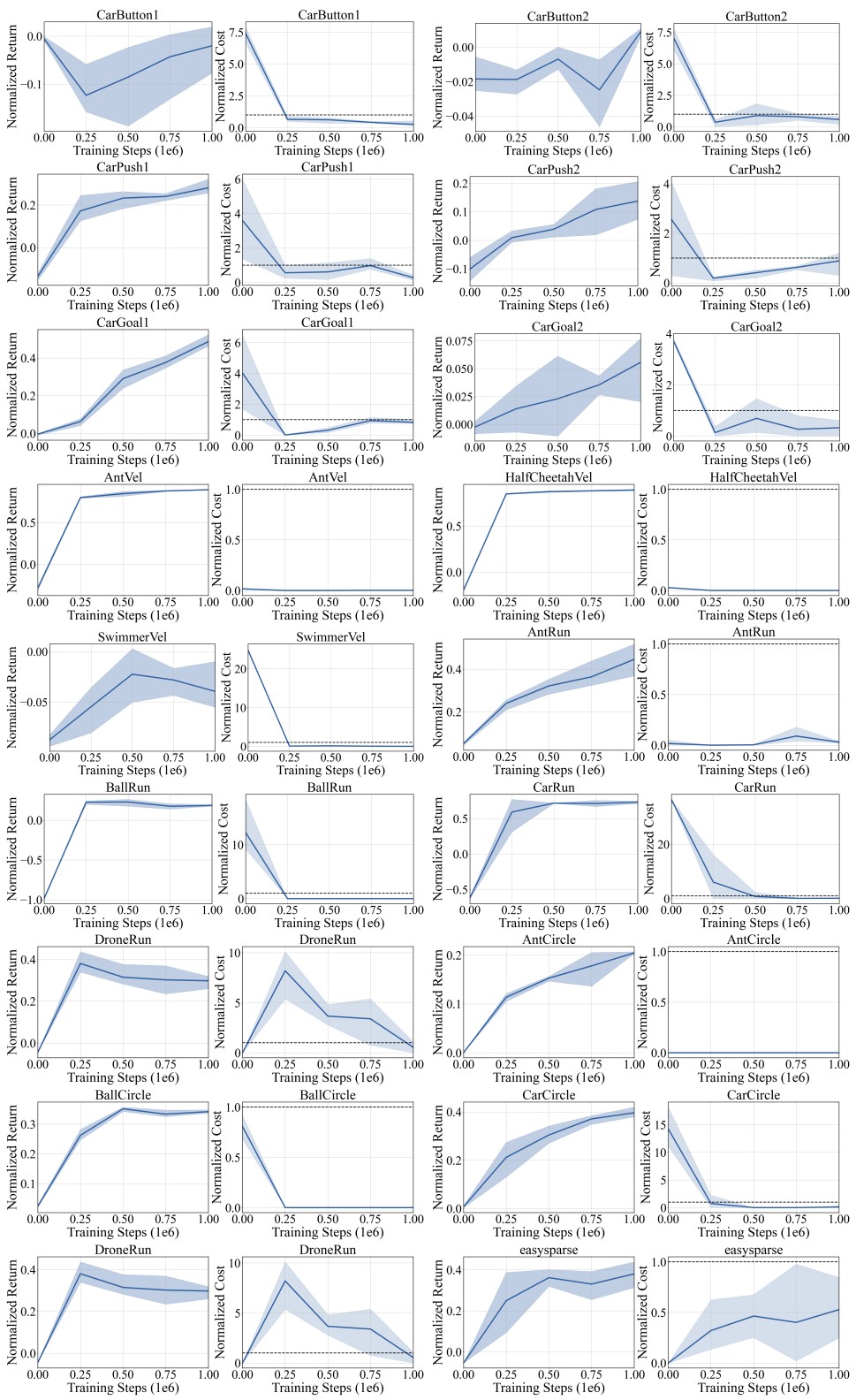

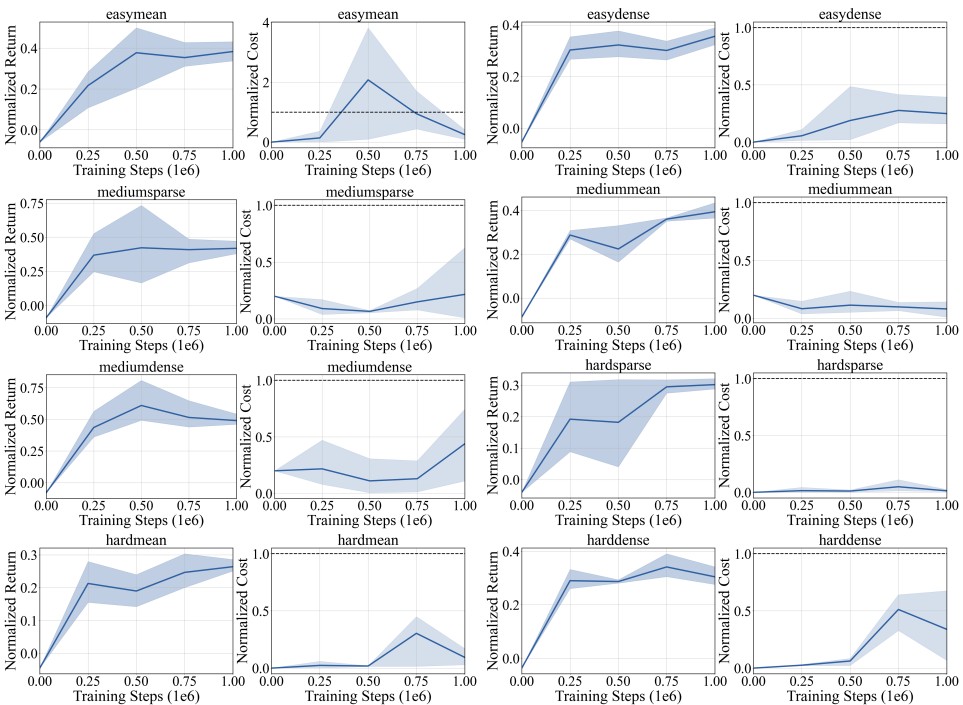

Figure 11: The learning curves for the 26 tasks in DSRL benchmark (Liu et al., 2023a).

