# OpenReview forum: "Safe Offline Reinforcement Learning with Feasibility-Guided Diffusion Model"
_ICLR.cc/2024/Conference — ICLR 2024 poster_

### Official Review · Reviewer_axZi · 2023-10-27

**Soundness:** 3 good
**Presentation:** 3 good
**Contribution:** 3 good
**Rating:** 8
**Confidence:** 3

**Summary:**

This paper proposes an algorithm for safe offline RL problems. Unlike most of the existing studies that deals with expected form of safety constraints, this paper deals with hard safety constraint (i.e., probability-1 constraints). The authors discover that the hard safety constraint can be equivalently translated to identifying the largest feasible region given the offline dataset, which leads to a feasibility-dependent objective which can be further solved using three simple decoupled learning objectives. The authors compare their proposed algorithm called FISOR against baselines on DSRL benchmark for safe offline RL.

**Strengths:**

- This paper is well-written and easy to follow. The problem settings are motivated and I think it is an important direction for safe RL research.
- The proposed method is solid and looks sound to me.
- The empirical evaluation are conducted in various benchmark tasks.

**Weaknesses:**

### Related Work

Hard constraints in safe RL have been studied in online RL settings. There are several important missing references as being represented by:

- Sootla, Aivar, et al. "Sauté rl: Almost surely safe reinforcement learning using state augmentation." International Conference on Machine Learning. PMLR, 2022.
- Wang, Yixuan, et al. "Enforcing hard constraints with soft barriers: Safe reinforcement learning in unknown stochastic environments." International Conference on Machine Learning. PMLR, 2023.
- Wachi, Akifumi, and Yanan Sui. "Safe reinforcement learning in constrained Markov decision processes." International Conference on Machine Learning. PMLR, 2020.

I agree with the authors that there is litte literature on safe **offline** RL and know that there is Table 4 in the Appendix. However, there are existing studies that are directly related to this paper. For example, Sootla et al. (2022) can be easily extended to the offline RL settings since all we have to do is to augment the state and provide a large penalty for violating the hard safety constraint.

### Experiments
While I would like to thank the authors for conducing good experiments, I am not fully convinced by the experimental results. First, all the baselines are based on the soft safety constraint and it seems unfair to simply compare their safety performances (Note that I do not say that such baseline methods should not be compared given there is few exising paper with the same problem settings). I personally consider that, if safe online RL algorithm can be extended to offline settings, such algorithms should be additionally included as a baseline. Second, I want the authors to discuss the experimental results on the reward performance. I understand sinec FISOR is enforced a hard constraint, it is natural that reward performance is deteriorated. However, under the condition that most of the baseline methods do not satisfy the safety constraint (because safety requirements are different by nature), I feel that it is rather arbitrary to compare the reward peformance **within** safe agents. Though the authors say that " FISOR is the only method that can
guarantee safety satisfaction in all tasks, while achieving top returns in most tasks", I personally feel that this claim is overstated. I think the experimental results are interesting; hence, I would like the authors to discuss more on the **Main Results** paragraph for future readers (e.g., why CPQ satisfy the safety constraints, why FISOR does not perform well in Safety Gym tasks, etc.)

**Questions:**

[Q1] I consider that there are several safe online RL algorithms with hard constraints. And some of them can be easily extended to offline RL settings. Could you give me some comments about my thoughts?

[Q2] In general, the performance of an offline RL algorithm largely depends on the data coverage of the offline dataset. Could you explain the (empirical) relationship of the performance if FISOR and the data coverage? Is FISOR more efficient in terms of the data coverage compared to other baselines?

[Q3] Why only CPQ satisfies the safety constraints within the baseline methods?

---

> ### Author Response · Authors · 2023-11-15
> **Response to Reviewer axZi**
>
> We thank the reviewer for the constructive comments and positive feedback on our paper. Regarding the concerns of the reviewer axZi, we provide the following responses:
> > **W1 & Q1 & W2.1 Some of safe online RL methods can be easily extended to offline RL settings, such algorithms should be included as a baseline.**
>
> We thank the reviewer for these works and we are happy to compare against them in our revised paper! We have added comparison with Sauté [1] and RCRL [2], since Sauté [1] is relatively versatile and RCRL [2] also adopts HJ Reachability to enforce hard constraints. Results can be found in Table 6, Appendix D.7 in our revised paper.
> - Note that applying safe online RL methods in offline settings faces distributional shifts. To address this, we add a BC constraint in Sauté [1] and RCRL [2] similar to an offline RL method TD3+BC [3] to combat distributional shifts. Also, we swept the conservatism strength to find a good balance between safety constraint and behavior constraint to obtain the best possible results.
>     - However, Table 6 shows that we cannot obtain good results with direct adaptations of safe online RL methods into offline settings, even with carefully tuned hyper-parameters, due to the intricate coupled training processes.
>     - FISOR, however, enjoys good results thanks to its decoupled learning objectives.
>
> Besides the experiments, we briefly discussed these safe online RL methods here:
> - For Sauté[1], similar to CPQ[4], safety constraints are handled by integrating them into the state space and modifying the objective accordingly. They struggle to handle zero cost limit, thus we still consider it to be a type of soft constraint method.
> - For [5], the intertwined learning process (barrier function, model, policy) is not suitable for offline scenarios.
> - For [6], its online safe exploration with environments is not compatible for offline settings.
>
> > **W2.2. Discuss the experimental results on the reward performance.**
> - The comparison criteria strictly follows the DSRL benchmark [7] (Q1 in general response), where safety is prioritized over rewards. We believe this makes sense as discussing the rewards is only meaningful when safety constraints are met in safety-critical tasks.
> - Previous methods may prioritize reward maximization while overlooking safety constraints. Thus, some baselines can obtain high rewards. Figure 3 shows that some baselines even encounter more safety violations than the ones trained without any safety constraint. This shows that existing safe offline RL methods are hard to strike the right balance between reward maximization and safety satisfaction. Thus, we didn't evaluate solely based on rewards, but filtered out the safe agents, and then compared the rewards, strictly following the DSRL [7]. We believe this evaluation protocol is more reasonable.
>
> > **W2.3. Discuss more on the Main Results**
> - Why does CPQ satisfy the safety constraints?
>     - CPQ is an over-conservative method that prioritizes safety rather than reward maximization. Specifically, CPQ harshly penalizes unsafe areas by pulling up the cost values on those regions using value penalty [8]. However, value penalty is proved over-conservative [9], making CPQ primarily focus on safety constraints. For example, CPQ obtains good safety results on MetaDrive tasks but performs poorly in terms of rewards.
>     - In addition, the offline value regularization is hard to tune [10], making CPQ perform not very well on other tasks.
> - Why FISOR does not perform well in Safety Gym tasks?
>     - No. Table 1 shows that FISOR meets safety requirements (normalized cost<1 according to the DSRL evaluation protocol) and meanwhile achieves competitive rewards on all SafetyGym tasks as compared to baselines. Moreover, FISOR has the lowest average cost scores as compared to all other baselines. It would be appreciated if the reviewer could tell us why the reviewer thinks FISOR does not perform well in Safety Gym tasks.
>
> > **Q2.  Could you explain the (empirical) relationship of the performance if FISOR and the data coverage? Is FISOR more efficient in terms of the data coverage compared to other baselines?**
> - We have added experiments on small data settings in Figure 10, Appendix D.8 in our revised paper. FISOR shows great stability across different dataset sizes, while other methods fail miserably under small data settings. We think the robustness in the low-data regime of FISOR is attributed to its stable decoupled learning procedure and policy extraction using weighted behavior cloning.
>
> > **Q3.  Why only CPQ satisfies the safety constraints within the baseline methods?**
>
> Please see the responses for W2.3 for details. Note that although being very conservative, CPQ still fails to satisfy safety constraints in many tasks as the value regularization of CPQ is hard to tune. We use bold black and blue to represent safe agents in Table 1. For specific evaluation protocol, please see the general response in Q1 and Q2.

---

> > ### Author Response · Authors · 2023-11-15
> > **References**
> >
> > [1] Sauté rl: Almost surely safe reinforcement learning using state augmentation. ICML 2022.
> >
> > [2] Reachability constrained reinforcement learning. ICML 2022.
> >
> > [3] A minimalist approach to offline reinforcement learning. NeurIPS 2021.
> >
> > [4] Constraints penalized q-learning for safe offline reinforcement learning. AAAI 2022.
> >
> > [5] Enforcing hard constraints with soft barriers: Safe reinforcement learning in unknown stochastic environments. ICML 2023.
> >
> > [6] Safe reinforcement learning in constrained Markov decision processes. ICML 2020.
> >
> > [7] Datasets and Benchmarks for Offline Safe Reinforcement Learning. 2023
> >
> > [8] Conservative q-learning for offline reinforcement learning. NeurIPS 2020.
> >
> > [9] Cal-ql: Calibrated offline rl pre-training for efficient online fine-tuning. 2023.
> >
> > [10] A minimalist approach to offline reinforcement learning. NeurIPS 2021.

---

> ### Comment · Reviewer_axZi · 2023-11-15
> **Thank you!!**
>
> I respectfully appreciate the authors for their clarifications and for adding new experiments. I've read the authors' feedback, other reviews, and revised manuscript.
>
> **W1 & Q1 & W2.1: Some safe online RL methods can be easily extended to offline RL settings, such algorithms should be included as a baseline.**
> - Thank you for adding informative experiments. I do appreciate the authors' effort in a short period. I am satisfied and convinced about this point.
>
> **W2.2, W2.3: Discuss the experimental results on the reward performance.**
> - Though I more or less agree with the authors, there is room to consider whether the authors' proposed method (FISOR) successfully solves the Safety-Gym environment given the low reward performance. I do agree that FISOR performs **better** than the baselines in most of the benchmark tasks, but there are several tasks that I personally think all algorithms could not solve. To put it in an extreme way, a very conservative policy (e.g., continuing to stay at the initial safe position) may be evaluated as a better policy if an agent does not have to get much reward.
>
> **Q3: Data coverage in offline RL settings.**
> - Thank you!! I think it is a nice addition for future readers.
>
> **W2.2 & Q3: Regarding CPQ**
> - Thank you for your explanation. I understand the reason why CPQ satisfies safety constraints.
>
> The authors have addressed most of my concerns at the time of the initial review, I increased my score from 6 to 8. I think this paper has been substantially improved. Thank you!!

---

> > ### Author Response · Authors · 2023-11-17
> > **Thank you!!**
> >
> > Dear reviewer axZi,
> >
> > Thank you so much for your detailed additional comments and for raising your score to 8!! We really appreciate it!! We rendered the policy rollouts of FISOR on some SafetyGym tasks that achieved low rewards. We observed the learned policy sometimes will deviate away from the goal to avoid obstacles, but the rewards are defined as punishing such deviation, such as the CarButton tasks as defined [linked here](https://www.safety-gymnasium.com/en/latest/environments/safe_navigation.html). In this case, the policy learned by FISOR will have negative rewards, but note that this policy is better than staying in the initial safe position since FISOR learns meaningful skills to move towards the goal and avoid unsafe regions. We will investigate this further to see if we can improve the rewards.
> >
> > Thanks again for these detailed suggestions!!

---

### Official Review · Reviewer_3zVo · 2023-11-01

**Soundness:** 4 excellent
**Presentation:** 3 good
**Contribution:** 4 excellent
**Rating:** 8
**Confidence:** 3

**Summary:**

This paper studies safe offline reinforcement learning by introducing feasibility into the RL problem for safety-critical systems. The authors convert the safe offline RL problem into a feasibility-dependent optimization problem, i.e., when feasible, maximize reward, when infeasible reduce the safety violation as much as possible.

**Strengths:**

1. The paper is novel and the idea is interesting. It scales the HJ reachability to the high-dimensional RL problem and leverages it to offline compute feasible regions, which helps build feasibility-dependent RL problems to encode hard constraints.
2. The paper is well-motivated, well-organized, and in general easy to follow.
3. The paper is sound, at least from the writing logic, given the fact I didn't check the math details in the appendix.
4. The experiments are comprehensive and the results are quite significant compared to the existing work.

**Weaknesses:**

1. The authors may need to add more references to cover "safe RL with hard constraints" to better motivate this work and claim the differences. A quick search gives me several recent papers, such as
a. Wang, Yixuan, et al. "Enforcing hard constraints with soft barriers: Safe reinforcement learning in unknown stochastic environments." International Conference on Machine Learning. PMLR, 2023.
And those in
b. Zhao, Weiye, et al. "State-wise safe reinforcement learning: A survey." arXiv preprint arXiv:2302.03122 (2023).

2. It seems that the baselines in the experiments all consider soft safety constraints. The reviewer would like to see the comparison with baselines belonging to safe offline RL with hard constraints if there is such work.

**Questions:**

1. because this work uses learning to learn feasible regions by HJ reachability, how "accurate" the learned regions are? How does the learning region error affect the optimization problem and safety violations?

---

> ### Author Response · Authors · 2023-11-15
> **Response to Reviewer 3zVo**
>
> We thank the reviewer for the constructive comments and positive feedback on our paper. Regarding the concerns of the reviewer 3zVo, we provide the following responses.
>
> > **W1. The authors may need to add more references to cover "safe RL with hard constraints" to better motivate this work and claim the differences.**
>
> We really appreciate the reviewer for providing these relevant works and we are more than happy to include them in our revised paper! Here, we briefly summarize the differences and relations with these related works:
>
> - For [1], the authors proposed a method for learning safe policy with joint soft barrier function learning, generative modeling, and policy optimization. Compared to HJ reachability, barrier functions may be overly conservative [2]. Also, the intertwined learning process is not suitable for offline RL scenarios.
> - For [3], although it considers state-wise constraints, this type of problem still allows for a certain degree of constraint violation. Therefore, we classify it as a soft constraint.
>
> Additionally, we have included more discussions of related works in Appendix A. Here, we provide a brief summarization.
>
> - For [4], although this method can achieve zero constraint violations, it requires domain knowledge of the task to design specific safety constraints, and thus is not suitable for general and more complex tasks.
> - For [5], the authors primarily focus on the expansion of safe regions in online exploration. However, interaction with the environment is not permitted in an offline setting. In contrast, we learn the feasible region directly from the dataset.
>
> > **W2. It seems that the baselines in the experiments all consider soft safety constraints.**
>
>  To the best of our knowledge, **FISOR is the first that considers hard constraints in the offline setting**. Please see Table 4 for details. This is why we only compared soft constraints in the safe offline benchmark DSRL [6].
> - The soft constraint baselines reported in our paper can be transformed into hard constraints by setting the cost limit to $0$. However, as shown in our paper (Figure 3), they suffer from extremely inaccurate estimations of the cost value and leads to increased safety violations (Table 2).
> - Some safe online RL methods that consider hard constraints can be extended to the offline RL settings by introducing some regularization to combat distributional shift. We combined RCRL [7], a safe online RL method that also uses HJ reachability, with TD3+BC [8] to adapt RCRL into the offline setting. The results can be found in Table 6, Appendix D.7 in our revised paper. Table 6 shows that RCRL+TD3BC fails miserably as it is hard to achieve a good balance between reward maximization, constraint satisfaction and distributional shift avoidance. FISOR, on the other hand, obtains great performance in the offline setting due to its decoupled learning procedure.
>
> > **Q1. How "accurate" the learned regions are? How does the learning region error affect the optimization problem and safety violations?**
>
> Please see Figure 1 for illustration. The learned feasible regions in FISOR are far more accurate than the ones learned by cost value functions.
> - In addition, we can select a small $0<\tau<1$ in $L_{rev}^{\tau}(u)$ of Eq.(8) to avoid infeasible states being mistakenly learned as feasible states. We have added demonstrative results in Figure 6, Appendix D.1 of the revised manuscript. Results show that a small $\tau$ will induce a smaller-sized learned feasible region, which is acceptable in safety-critical scenarios as it will induce safer policies. Figure 4(a) also supports this argument.
> - Moreover, estimation errors may increase as the amount of offline data decreases. We have added experiments on small dataset settings in Figure 10, Appendix D.8. Results show that FISOR is robust to small dataset settings as compared to baselines, thanks to its great training stability.
>
>
> [1] Enforcing hard constraints with soft barriers: Safe reinforcement learning in unknown stochastic environments. ICML 2023.
>
> [2] Lyapunov density models: Constraining distribution shift in learning-based control. ICML 2022.
>
> [3] State-wise safe reinforcement learning: A survey. 2023
>
> [4] Model-free safe control for zero-violation reinforcement learning. CoRL 2021.
>
> [5] Safe reinforcement learning in constrained Markov decision processes. ICML 2020.
>
> [6] Datasets and Benchmarks for Offline Safe Reinforcement Learning. 2023
>
> [7] Reachability constrained reinforcement learning. ICML 2022.
>
> [8] A minimalist approach to offline reinforcement learning. NeurIPS 2021.

---

> > ### Comment · Reviewer_3zVo · 2023-11-16
> >
> > I appreciate the feedback from the authors. The responses have fully answered my questions. I am happy to raise the value to an 8 and hope to see the paper get accepted.

---

> > > ### Author Response · Authors · 2023-11-17
> > > **Thanks for increasing to 8!**
> > >
> > > Dear reviewer 3zVo,
> > >
> > > Thank you so much for your effort engaged in the review phase and for increasing to 8! We're more than happy to know that we have fully addressed your questions! Thanks again!

---

### Official Review · Reviewer_FU8i · 2023-11-02

**Soundness:** 2 fair
**Presentation:** 3 good
**Contribution:** 3 good
**Rating:** 6
**Confidence:** 3

**Summary:**

Safe offline reinforcement learning aims to develop safe policies without risky online interactions, but most existing methods risk potential safety violations by only enforcing soft constraints. This work presents a new approach, FISOR (FeasIbility-guided Safe Offline RL), utilizing reachability analysis from safe-control theory to enforce hard safety constraints, by maximizing rewards within a feasible region determined by the offline dataset and minimizing risks elsewhere. FISOR decouples the process into safety adherence and reward maximization. The result outperforms baselines in safety satisfaction and return performance in the DSRL benchmark for safe offline RL.

**Strengths:**

- The idea of including reachability analysis into the learning for safety satification is interesting.
- The paper is well-written.

**Weaknesses:**

- There seems to be an inconsistency between the goal and the actual approach. The paper aims for 100% guarantee in safety. However, the approach still need to estimate the feasible areas and then to yield the value function threshold. There could be estimation error and this estimation requires very thorough dataset provided, which may not be practical in reality.
- The learnt infeasible states may be false negative. How do you assure that the infeasible states are really unsolvable instead of you able not able to solve it?

**Questions:**

- How do you guarantee the gap between the ground-truth infeasible regions and the learnt infeasible regions?
- How do you measure the normalized cost in the evaluation? Why is it not zero for cases that are 100% safe?

---

> ### Author Response · Authors · 2023-11-15
> **Response to Reviewer FU8i**
>
> We thank the reviewer for the constructive comments. Regarding the concerns of the reviewer FU8i, we provide the following responses.
>
> > **W1. There seems to be an inconsistency between the goal and the actual approach. The paper aims for 100% guarantee safety. However, the approach still needs to estimate the feasible areas and then yield the value function threshold. There could be estimation error and this estimation requires very thorough dataset provided, which may not be practical in reality.**
>
> We agree with the reviewer that we use deep neural networks to estimate the feasible regions and approximation errors could exist. However, we must clarify the following major misunderstandings.
>
> - First, note that the feasible areas assessed using HJ reachability are **far more accurate and robust** as compared to those measured using cost value functions, which are widely used in previous works, as demonstrated in Figure 1. Therefore, introducing HJ reachability into safe offline RL contributes a major step towards ensuring safety in safe offline RL settings.
> - Moreover, It can be quite challenging to accurately specify feasible regions in high-dimensional continuous control tasks without functional approximators such as deep neural networks. For example, MetaDrive tasks have extremely complex state spaces with over 250 dimensions. In this sense, it is **impractical** to enforce hard constraints without functional approximators. Specifically, previous safe online RL methods, that aim for 100% safety guarantee, also require to estimate the feasible regions, and approximation errors also exist [1][2][3]. None of these methods can achieve the ideal 100% safety guarantee unless the full state-action space is fully explored.
>
> To further address the reviewer's concern, we evaluated FISOR in small dataset settings. Please see Figure 10, Appendix D.8 for details.
>
> - Briefly, Figure 10 shows that a small offline dataset doesn't greatly impact FISOR's safety or performance. **This shows that FISOR is robust for small-sample scenarios**.
> - In addition, FISOR is the first offline RL method that considers hard constraints and proposes a stable decoupled training procedure. As shown in Table 1, FISOR is the only method that can ensure safety in all 26 benchmark tasks, while maintaining the highest reward in most of the tasks as compared to other safe baseline agents. This is a remarkable result since the safe offline RL problem with hard constraint satisfaction itself is an extremely challenging task, as it requires seeking the tricky balance among reward maximization, safety constraints satisfaction and distributional shift avoidance.
>
>
> > **W2. The learned infeasible states may be false negative. How do you assure that the infeasible states are really unsolvable?**
>
> Controlling $\tau$ in Eq. (8) adjusts the feasible region's conservatism. A higher $\tau$ near 1 means learning near-maximum feasible regions, while lowering $\tau$ leads to a smaller feasible region.
>
> - We added the visualization of learned feasible regions using different $\tau$ values in Figure 6, Appendix D.1 in our revised paper. Figure 4(a) also supports this argument.
> - In practice, we select $\tau=0.9<1$, from which a smaller feasible region will be learned and enforces harder safety constraints, where small errors will not likely lead to false negative but false positive. However, a false positive is acceptable since it will lead to a more conservative and safe policy. Although this may lead to some reward loss, it is acceptable in safety-critical scenarios.
> - If false negative leads to unsatisfactory unsafety by chance, we can increase the algorithm's conservatism by reducing the value of $\tau$.
>
> > **Q1. How do you guarantee the gap between the ground-truth infeasible regions and the learnt infeasible regions?**
>
> Please see the responses to W2 for details.
>
> > **Q2.1. How do you measure the normalized cost in the evaluation? Why is it not zero for cases that are 100% safe?**
>
> For the calculation method of normalized cost, please see the general response Q1. We respond to the latter question in W1.
>
>
> [1] Reinforcement Learning for Safety-Critical Control under Model Uncertainty, using Control Lyapunov Functions and Control Barrier Functions. RSS 2020.
>
> [2] Model-free safe reinforcement learning through neural barrier certificate. RAL 2023
>
> [3] Enforcing hard constraints with soft barriers: Safe reinforcement learning in unknown stochastic environments. ICML 2023.

---

> > ### Comment · Reviewer_FU8i · 2023-11-16
> > **Thank you for your clarification.**
> >
> > I appreciate the authors for their thorough reply. My major concerns have been addressed. Thus, I increased the score to 6.

---

> > > ### Author Response · Authors · 2023-11-17
> > > **Thanks for increasing to a 6!**
> > >
> > > Dear reviewer FU8i,
> > >
> > > Thanks for your effort in the review phase and for increasing to 6! We're happy to know that we have addressed your major concerns! Thanks again!

---

### Official Review · Reviewer_5fk7 · 2023-11-02

**Soundness:** 3 good
**Presentation:** 3 good
**Contribution:** 3 good
**Rating:** 8
**Confidence:** 2

**Summary:**

The paper introduces a method for safe offline policy learning with hard constraints, called FISOR. The method works through three dedicated learning objects that separate the feasible and infeasible region plus satisfaction of safety constraints. Policy learning is performed via a guided diffusion model and experiments show that FISOR is both effective int terms of high rewards and safety.

**Strengths:**

Safety in RL is an important and relevant topic and this paper is a nice addition to the existing body of work. The approach to introduce safety via the three learning objectives is well motivated and substantially explained (although I did not follow all details). The usage of the diffusion model for policy learning is interesting.

Extensive experiments including ablation studies are performed on well-known environments.
While FISOR does not achieve the highest rewards in these environments, it consistently is safe and has lower cost than the baseline methods, except for CPQ in the metadrive environment.

**Weaknesses:**

The presentation of the results in Table 1 is a bit confusing. What is the criterion for an agent to be safe? Is it formulated as a threshold on the cost or do you count actual safety violations and only indicate those?

I would like a more thorough discussions of limitations of the method. In the conclusion the limited availability of offline RL data is mentioned, but it is (if I didn't miss something) not well motivated or shown how this limitation affects the method or if it is more a general concern.
What are other limitations? Are there safety constraints that cannot be handled or that are substantially more difficult?

The figures are a bit blurry on my reader, maybe they can be converted to a different format.

**Questions:**

Table 1: What is the criterion for an agent to be safe? Is it formulated as a threshold on the cost or do you count actual safety violations and only indicate those?

What are other limitations?

Are there safety constraints that cannot be handled or that are substantially more difficult?

---

> ### Author Response · Authors · 2023-11-15
> **Response to Reviewer 5fk7**
>
> We thank the reviewer for the constructive comments and positive feedback on our paper. Regarding the concerns from the reviewer 5fk7, we provide the detailed responses separately as follows:
>
> > **W1. & Q1. What is the criterion for an agent to be safe? Is it formulated as a threshold on the cost or do you count actual safety violations and only indicate those?**
>
> - Please refer to Q1 and Q2 in general response for details about our evaluation protocol.
>
> > **W2.1. How does limited data affect the method ?**
>
> - Limited data may have negative impacts on the performances, which is a common issue with existing offline algorithms, as high-performance offline policy learning usually requires a reasonable amount of offline training data as well as good state-action space coverage.
> - We have added new experiments on small data settings in Figure 10, Appendix D.8 in our revised paper. Figure 10 shows that FISOR can still maintain satisfactory safety levels with limited data, with a lower level of safety performance degeneration as compared to other methods. This could be attributed to the decoupled training approach in FISOR, which enhances policy learning stability under a low-data regime.
>
> > **W2.2. & Q2. What are other limitations?**
>
> Besides the limitation mentioned in the conclusion, please see Appendix F for more discussions on the potential limitations of FISOR. For other potential limitations:
> - FISOR contains four hyper-parameters, such as the forward/reverse expectile and policy extraction temperatures. However, we find that FISOR is not very sensitive to hyper-parameter changes and is robust to different tasks with the same hyper-parameters. We used **only one set** of hyper-parameters ($\tau=0.9$ for both forward/reverse expectile value), $\alpha_1=3, \alpha_2=5$) and achieved consistently good safety performances and reward returns among 26 tasks, as shown in Table 1.
>
> > **W2.3. & Q3. Are there safety constraints that cannot be handled or that are substantially more difficult?**
>
> We appreciate the reviewer for the constructive comment. We have added the following limitation in Appendix F.
> - In our setting, FISOR considers a very general form of constraint, which sets the safety violation signal to $0$ if safe and $1$ if unsafe. This allows flexible extension to a wide range of hard constraint satisfaction problems. However, this setup may not be applicable to constraints involving environmental disturbances [1] or probabilistic constraints [2]. These safety constraints might impact cost value estimations, thereby affecting the performance of the algorithm.
>
> > **W3. The figures are a bit blurry.**
>
> We thank the reviewer for this kind comment. We have increased the resolution of the images. We would really appreciate it if the reviewer could provide more specific suggestions so that we can make further improvements.
>
> [1] Safe reinforcement learning using robust control barrier functions. RAL 2022.
>
> [2] Separated proportional-integral lagrangian for chance constrained reinforcement learning. IEEE Intelligent Vehicles Symposium 2021.

---

> > ### Comment · Reviewer_5fk7 · 2023-11-16
> >
> > Thank you for your reply and answering my questions/concerns.
> >
> > The answers (and the other responses) take into account my criticism and the revision of the paper has also improved it.
> > This is a good paper and I hope to see it accepted.

---

> > > ### Author Response · Authors · 2023-11-17
> > > **Thanks for the positive feedback on our work!**
> > >
> > > Dear reviewer 5fk7,
> > >
> > > Thank you for the effort engaged in the review phase and your positive feedback on our work! It's great to know that we have addressed your concerns! Thank you!

---

### Author Response · Authors · 2023-11-15
**General Response**

We thank all the reviewers for the effort engaged in the review phase and the constructive comments. Regarding the common concerns of the reviewers, we provide the following responses.

> **Q1. Evaluation protocol in benchmark tasks:**

- Our evaluation protocol strictly follows the recent safe offline RL benchmark DSRL [1], we use the normalized reward return and the normalized cost returns as the evaluation metrics:
    $R_{\rm normalized} = \frac{R_{\pi}-r_{\rm min}}{r_{\rm max}-r_{\rm min}}$, $C_{\rm normalized} = \frac{C_{\pi}+\epsilon}{l+\epsilon}$, where $R_{\pi}=\sum_t r_t$ is the policy's reward return and $C_{\pi}=\sum_t c_t$ is its cost return. In all the benchmark tasks, the per-step cost $c_t=0$ if the state is safe and $c_t>0$ for unsafe states [2][3][4]. $r_{\rm min}$ and $r_{\rm max}$ are the minimum and maximum reward return in the offline data, respectively. $l$ is a human-defined cost limit. $\epsilon$ is a positive number to ensure numerical stability if $l=0$.
- Following the DSRL[1] benchmark, the task is considered safe when $C_{\pi}$ does not exceed the cost limit $l$, or in other words $C_{\rm normalized}$ is no larger than 1.

> **Q2. Comparison criteria on the performance of evaluated methods:**

- We follow the DSRL[1] benchmark's evaluation protocol, where safety is prioritized over rewards, same as the common consideration in typical safety-critical scenarios. For safe agents ($C_{\rm normalized} ≤ 1$), we compare them using normalized reward returns. The one with the highest $R_{\rm normalized}$ is considered the superior agent. Those not meeting safety constraints are considered as unsafe agents ($C_{\rm normalized} > 1$).

[1] Datasets and Benchmarks for Offline Safe Reinforcement Learning. 2023.

[2] Benchmarking safe exploration in deep reinforcement learning. 2019.

[3] Bullet-safety-gym: A framework for constrained reinforcement learning. 2022.

[4] Metadrive: Composing diverse driving scenarios for generalizable reinforcement learning. IEEE transactions on pattern analysis and machine intelligence 2022.

**We have revised our paper (highlighted in blue text color). The modifications are summarized as follows.**

1. (For Reviewer 3zVo, axZi) We added more references about hard constraints, and further discussions are provided in Appendix A.
2. (For Reviewer FU8i, 3zVo) We added the visualization of learned feasible regions using different $\tau$ values in Figure 6, Appendix D.1.
3. (For Reviewer 5fk7, FU8i, axZi) We added detailed benchmarks' evaluation metrics in Appendix D.3.
4. (For Reviewer 3zVo, axZi) We added experiments of extending safe online RL to offline settings in Table 6, Appendix D.7.
5. (For Reviewer 5fk7, FU8i, 3zVo, axZi) We added experiments of small datasets in Figure 10, Appendix D.8.
6. (For Reviewer 5fk7) We added more limitation discussions in Appendix F.

---

### Meta-Review · Area_Chair_7D8X · 2023-12-05

**Metareview:**

to be completed

**Justification For Why Not Higher Score:**

Based on deep neural networks, the method remains empirical.  There can still be errors in the estimation of the feasible regions.

**Justification For Why Not Lower Score:**

This paper addresses an important problem.  It is well written, and the new method is both well motivated and interesting.  Extensive experiments including ablations support the good performance of the method.  All reviewers and myself find the paper a good addition to the conference.

---

### Decision · Program_Chairs · 2024-01-16

Accept (poster)